# First Results of Cloud Retrieval from Geostationary Environmental Monitoring Spectrometer

Bo-Ram Kim [1,2], Gyuyeon Kim [3], Minjeong Cho [3], Yong-Sang Choi [3, *], and Jhoon Kim [4]

[1]Satellite Application Division, National Satellite Operation and Application Center, Korea Aerospace Research Institute, Daejeon 34133, Republic of Korea
[2]Department of Atmospheric Science and Engineering, Ewha Womans University, Seoul 03760, Republic of Korea
[3]Department of Climate and Energy Systems Engineering, Ewha Womans University, Seoul 03760, Republic of Korea
[4]Department of Atmospheric Science, Yonsei University, Seoul 03722, Republic of Korea

*Correspondence to*: Yong-Sang Choi (ysc@ewha.ac.kr)

**Abstract.** This paper introduces a cloud retrieval algorithm for the Geostationary Environmental Monitoring Spectrometer (GEMS), the first environmental geostationary orbit satellite, and validates its cloud products by comparing them with those produced by other instruments (OMI, TROPOMI, AMI, and CALIOP). The GEMS cloud products are corrected for the impact of clouds on the retrieval of atmospheric components, using the $O_2$-$O_2$ absorption band to retrieve the effective cloud fraction and cloud centroid pressure. The performance of the GEMS retrieval algorithm was similar to that of the OMI instrument. We analyzed the cloud retrieval characteristics for air pollution, typhoons, and sea fog in the East Asia region to evaluate whether GEMS cloud data can accurately represent various cloud features. Also, we evaluated the accuracy of the cloud retrieval algorithm through monthly validation for 2 years. The validation results provide a basis for future improvements of the GEMS cloud retrieval algorithm.

## 1 Introduction

Atmospheric composition has been monitored continuously by several satellite-loaded instruments since 1978: Total Ozone Mapping Spectrometer (TOMS), the Ozone Monitoring Instrument (OMI), Global Ozone Monitoring Experiment (GOME), SCanning Imaging Absorption spectroMeter for Atmospheric CHartographY (SCIAMACHY), and Tropospheric Monitoring Instrument (TROPOMI) (Hsu et al., 1997; Burrows et al., 1999; Bovensmann et al., 1999; Levelt et al., 2006; Veefkind et al., 2012). These spectrometers measure the ultraviolet (UV) and visible (Vis) radiation centered at 240 to 790 nm (Hsu et al., 1997; Burrows et al., 1999; Bovensmann et al., 1999; Levelt et al., 2006; Veefkind et al., 2012). It is then required to estimate the beam path length of the radiation to retrieve precise atmospheric compositions from the radiation measured by these spectrometers. The beam path length of the radiation is the entire path length of incoming and reflected solar energy by Earth's surface until reaching the satellite. Thus, the calculation requires to consider geometric factors such as solar zenith angle (SZA) and viewing zenith angle (VZA).

The beam path length should be also calculated not only for a clear-sky condition, but also for a cloudy-sky condition. This is because cloud layers can shorten the beam path length by blocking the beam from atmospheric components below the clouds. Cloud reflectance is typically greater than that of most surfaces (excluding snow and ice) and this cloud effect can inevitably result in significant errors in the observations of atmospheric variables (Hong et al., 2017; Chimot et al., 2018). Therefore, to obtain accurate concentrations of atmospheric components, it is necessary to evaluate and quantify the cloud effects on the beam path length.

A cloud can exhibit significant spatiotemporal variability, and its characteristics can vary widely depending on the instrument. To accurately measure atmospheric components, it is first necessary to obtain real-time cloud information that reflects the characteristics of the satellite's instruments. Instruments that monitor the atmospheric environment use their own algorithms to extract information on clouds. In previous studies, spectral bands that reveal phenomena involving gases with relatively constant proportions in the dry atmosphere, such as those associated with rotational Raman scattering (RRS) (Joiner et al., 2006; Vasilkov et al., 2008), $O_2$-$O_2$ absorption (Acarreta et al., 2004; Stammes et al., 2008; Veefkind et al., 2016; Vasilkov et al., 2018), and $O_2$-A absorption (Wang et al., 2008; Wang et al., 2014; Loyola et al., 2018; Compernolle et al., 2021; Yang et al., 2021; Taylor et al., 2021), have been used for cloud retrieval. The GOME, SCIAMACHY, and TROPOMI instruments use the $O_2$-A band to retrieve cloud information, while the OMI (without the $O_2$-A band) retrieves information using two methods: Raman scattering and the $O_2$-$O_2$ absorption band (Acarreta et al., 2004; Joiner et al., 2006; Stammes et al., 2008; Vasilkov et al., 2008; Wang et al., 2008; Vasilkov et al., 2018; Veefkind et al., 2016; Wang et al., 2018; Loyola et al., 2018; Compernolle et al., 2021). The Greenhouse Gases Observing Satelltie (GOSAT) and the greenhouse gas (GHG)Sat instruments make use of the $O_2$-A band to monitor clouds and aerosols, perform cloud screening, and make cloud-cover estimates to gauge the influence of clouds. The OCO-2 satellite also monitors the $O_2$-A band, which is used for cloud screening (Taylor et al., 2011; Yang et al., 2021).

GEMS, was launched in 2020, was the first environmental monitoring satellite in geostationary earth orbit (GEO) (Kim et al., 2020). GEMS measures levels of ozone, aerosols, nitrogen dioxide, sulfur dioxide, formaldehyde, and other atmospheric constituents in the UV-Vis range over East Asia. It monitors the 300–500 nm range with a spectral resolution of 0.6 nm. As it operates in GEO, it can monitor the same field of view every hour. Previous satellites for atmospheric environmental monitoring operated primarily in a sun-synchronous orbit (SSO) while maintaining the SZA variation, whereas GEMS operates in GEO, making it essential to consider large variations in SZA and VZA. The SZA varies greatly from dawn to noon and to sunset. While satellites in SSO perform observations close to noon to take of advantage of maximum illumination of Earth, a GEO satellite operates throughout the day, from dawn to sunset. This reduces the quantity of radiation energy reaching the satellite and extends the beam path, resulting in a lower signal-to-noise ratio (Vandaele et al., 2018) and cloud retrieval errors. The development and evaluation of algorithms that take these factors into account are crucial for accurate retrieval of cloud data (Kim et al., 2021a). Similarly, due to the wider observation range of the VZA compared to that possible in a low Earth orbit, errors may occur due to the increase in beam path length and the increase of the VZA. As this was the first time an environmental monitoring satellite was to be operated from a GEO, previous studies were conducted to consider the

observational characteristics due to differences in orbit (Kwon et al., 2017; Kim et al., 2018; Bak et al., 2019; Go et al., 2020; Kim et al., 2021a).

Here, we introduce a cloud retrieval algorithm for GEMS that takes into account the characteristics of observations made from GEO, and analyze the cloud properties retrieved from GEMS data. To ensure rapid and stable operation, the GEMS cloud retrieval algorithm was designed based on a look-up table (LUT) for the $O_2$-$O_2$ absorption band. The LUT was created

assuming the range of values that can be observed by GEMS-specific conditions for variables that affect $O_2$-$O_2$ absorption, such as SZA, VZA, surface reflectance, and surface pressure. A detailed description of the algorithm is provided in Section 2. To verify the performance of the algorithm, data from the OMI and TROPOMI instruments were used, and the Advanced Meteorological Imager (AMI) and Cloud-Aerosol Lidar with Orthogonal Polarization (CALIOP) cloud products were used to validate the GEMS products. The data are introduced in Section 3, and the validation results are shown and discussed in Section

4. Section 5 provides a summary and overall evaluation of the GEMS cloud retrieval algorithm.

## 2 Geostationary Environmental Monitoring Spectrometer (GEMS) Cloud Retrieval Algorithm

The GEMS instrument is aboard the Geostationary Korea Multipurpose Satellite2B (GK2B), a geostationary satellite launched by South Korea in 2020 and orbiting at 128.2°E to provide continuous atmospheric observation of East Asia. It is the first geostationary satellite to monitor atmospheric conditions in the 300–500 nm range at a resolution of 0.6 nm. As a UV-Vis

observation instrument, GEMS observes the East Asian region (5°S to 45°N, 75°E to 145°E) at 1-hour intervals during the daytime hours (0000-0900, 2200-2300 UTC) (Kim et al., 2020). The atmospheric components that GEMS measures, such as ozone, nitrogen dioxide, sulfur dioxide, formaldehyde, glyoxal, and aerosols, can be affected by clouds, leading to reduced accuracy and even making some components impossible to detect. This section introduces a cloud retrieval algorithm (Section 2.1), and defines the characteristics of clouds that can cause errors in estimating the concentration of atmospheric components

(Section 2.2).

### 2.1 GEMS Cloud Retrieval Algorithm Description

In the GEMS cloud retrieval algorithm, the observed radiance (RAD) and irradiance (IRR) values, as well as the observation geometry information, including SZA, VZA, and relative azimuth angle (RAA), and surface information such as surface pressure and surface reflectance were used as input values. Before the launch of GEMS, we designed the algorithm using 460–

490 nm. However, we discovered that bad-pixels exist over 485 nm in partial area (Lee et al., 2023) during the in-orbit test of GEMS. Therefore, the fitting window was shortened for stable cloud retrieval. To account for this issue that occurs in some observation areas of GEMS, the range of wavelengths for the input data in the fitting method was adjusted to 460–485 nm. Cloud detection is based on the reflection and scattering properties of clouds, making it possible to determine the presence of clouds throughout the GEMS observational spectrum. Cloud pressure is quantified by changes in the absorption or filling

phenomena due to variations in the beam path length caused by clouds. However, quantifying the impact of clouds when the

concentration of gases can undergo significant changes can be challenging, and it is essential to select areas with minimal variation in concentrations of trace gases.

The GEMS cloud retrieval algorithm assumes the mixed Lambertian equivalence reflectivity (MLER) cloud model, which is the model assumption used in the OMI. Similarly, for $O_2$-$O_2$ band–based cloud retrieval by OMI, the algorithm runs a radiative transfer model with inputs of observation geometry, surface information, effective cloud fraction (ECF), and cloud centroid pressure (CCP). The algorithm uses a simulated result with a 0.6 nm interval in the 460–485 nm range to create an LUT. Detailed input values for the LUT are presented in Table 1. While OMI defines nodal points for observation geometry based on the angle spacing, GEMS defines nodal points for the LUT based on the cosine value of observation geometry to ensure a linear relationship with beam path length. This approach allows for a simulation of the LUT with a superior linear interpolation scheme. For the radiative transfer simulation used to create the LUT, the Vector LInearized Discrete Ordinate Radiative Transfer (VLIDORT) NGST version (Spurr et al., 2006) was applied and the atmospheric profile as assumed based on the profiles of the Deriving Information on Surface conditions from Column and Vertically Resolved Observations Relevant to Air Quality (DISCOVER-AQ) campaign (Flynn et al., 2014). The simulated radiance in the 460–485 nm range was converted to reflectance assuming the observed reflectance ($R_\lambda$) in Eq. (1) is the ratio between the simulated RAD and IRR.

$$R_\lambda = \frac{\pi \cdot RAD}{IRR \cdot \cos SZA} \tag{1}$$

The observed reflectance spectrum with a 0.6 nm interval in the 460–485 nm range are used as input data for the fitting method based on the Differential Optical Absorption Spectroscopy (DOAS) method (Platt et al., 1979), as described by Eq. (2), which quantifies $O_2$-$O_2$ absorption.

$$ln(-R_\lambda) = C_1 + C_2 \times \lambda + N_{s,O_2-O_2} \times \sigma_{O_2-O_2} + N_{s,O_3} \times \sigma_{O_3} \tag{2}$$

where $R_\lambda$ represents the 43 input observed reflectance spectra, $\lambda$ is the wavelength for the 43 reflectance spectra, and $C_1$ and $C_2$ are the offset and slope of the linear component, respectively. $N_s$ represents the slant column density (SCD), and $\sigma$ refers to the absorption coefficient.

In this study, because only the effects of $O_2$-$O_2$ and $O_3$ absorption were considered (Brion et al., 1998; Thalman et al., 2013), only the column densities for $O_2$-$O_2$ ($N_{s,\ O2-O2}$) and $O_3$ ($N_{s,\ O3}$) absorption were calculated. The DOAS method separates the contribution of each gas's absorption coefficient as a weighting factor in the regions where absorption occurs, providing a linear element and the SCD of each absorption coefficient from the input reflectance spectrum. In addition, even though $NO_2$ absorption coefficients exist in the spectral range of the input reflectance, their effects are disregarded because the impact is negligible. In contrast, because the absorption coefficient of ozone has a peak near 480 nm, and its absorption line is prominent in the case of upper-level clouds, affecting the quantification of $O_2$-$O_2$ absorption, it should be removed.

The LUT was constructed based on two intermediate products The first is the continuum reflectance (CR, $R_c$), which is obtained by calculating the values of linear elements obtained through DOAS using Eq. (3).

$$R_c = C_1 + C_2 \times \lambda, \lambda = 477\ nm \tag{3}$$

The second intermediate product was based on the O$_2$-O$_2$ SCD values obtained through DOAS. In the GEMS cloud retrieval algorithm (Fig. 1), when the observed reflectance spectrum is provided, the intermediate products ($R_c$ and $N_{s,\ O2\text{-}O2}$) are first calculated via DOAS. These intermediate products are then fed into the LUT, along with observation geometry and surface information. Through a linear interpolation process, the final products (ECF and CCP) are produced.

Although the GEMS cloud retrieval algorithm is generally similar to the OMI cloud retrieval algorithm, significant differences are in the process of generating an LUT. As already noted, the observational geometry differs from the LEO orbit because GEMS is in GEO. The VZA is constant at each location, but it varies significantly across the observation area. In contrast, SZA varies significantly across the complete observation area from sunrise to sunset. This change in SZA influences not only the calculation of reflectance, the optical path, etc., but also the variation in surface reflectance. Therefore, we simulated the radiative transfer model (RTM) considering the various conditions of surface reflectance. To minimize the error of linear interpolation during the algorithm process, we defined the LUT using the cosine values of the zenith angle. In addition, different models were used to generate the LUT for the two algorithms. OMI used the Doubling-Adding-KNMI (DAK) model (Acarreta et al., 2004; Stammes et al., 2008; Veefkind et al., 2016; Vailkov et al., 2018), while GEMS used the VLIDORT NGST version (Spurr et al., 2006).

## 2.2 GEMS Cloud Products: Effective Cloud Fraction and Cloud Centroid Pressure

The visible and infrared regions of the electromagnetic spectrum have been commonly used for cloud-property retrieval, primarily from weather satellites. It can be challenging to measure cloud properties, such as top pressure and phase, in the UV-Vis region (Compernolle et al., 2021; Kim et al., 2019). In the GEMS cloud retrieval algorithm, cloud products are defined and produced to correct for the influence of clouds in other GEMS gaseous products over a limited spectral range. The GEMS spectral range is limited to those required to monitor specific cloud properties. We assumed that the cloud is a simple reflective surface, rather than apply a mie scattering model that takes into account cloud phase and effective radius.

In the GEMS spectrum, clouds are more reflective than the Earth's surface, with the exception of snow- and ice-covered terrain, and are typically at higher altitudes than the air pollutants that GEMS is measuring. Cloud products for estimating atmospheric components should correct for increased reflectance and beam path length. For the GEMS cloud retrieval algorithm, cloud products can be defined by two parameters: ECF and CCP, which are also used in the O$_2$-O$_2$ band algorithm (OMCLDO2) of OMI (Acarreta et al., 2004; Stammes et al., 2008; Veefkind et al., 2016; Vasilkov et al., 2018).

The ECF is calculated (Eq. 1) by assuming a Lambertian equivalent reflectivity (LER) of 0.8 for clouds (Accareta et al., 2004; Stammes et al., 2008), and comparing the observed reflectance ($R_{obs}$) at a pixel with the simulated reflectance under clear-sky ($R_{clr}$) and cloudy-sky ($R_{cld}$) conditions.

$$ECF = \frac{R_{obs} - R_{clr}}{R_{cld} - R_{clr}} \tag{3}$$

An LER cloud model with a reflectivity of 0.8 is assumed to represent a sufficiently thick cloud with an optical thickness of approximately 30, and the effects of reflection within the cloud and multiple scattering that occurs beneath the cloud are

ignored. A cloud model that defines the radiance of observed pixels by combining clear-sky and cloudy-sky pixels, assuming a reflectance of 0.8 for clouds and independent relationships between the two types of pixels (independent pixel approximation [Acarreta et al., 2004; Stammes et al., 2008; Veefkind et al., 2016; Vasilkov et al., 2018]), is an MLER model. The pressure at which reflection occurs due to the cloud is defined as the CCP. In addition, a cloud radiative fraction (CRF) for the wavelengths at which the atmospheric components are retrieved, is defined as a ratio of the radiance reflected by clouds to the

observed radiance for the atmospheric components. Although GEMS produces three types of cloud products, this study compares only the ECF and CCP for verification with other satellites, excluding the CRF.

## 3 Data and Method

Four satellite datasets (Table 2) used to validate the GEMS cloud retrieval algorithm are introduced in Section 3.1 and collocation methods for comparison of each satellite's data are presented in Section 3.2. To evaluate the performance of the

GEMS cloud retrieval algorithm, cloud products were produced using OMI radiance data, which are similar to the GEMS spectral resolution and cloud prototype algorithm. TROPOMI, an environmental satellite in operational since 2018, was also used to analyze the characteristics of clouds that appear during GEMS operations. AMI cloud data were gathered from the same GEO orbit but with a different definition of clouds. Also, CALIOP data were used for validation that based on the vertical distribution of clouds using an active sensor.

### 3.1 Satellite Data

### 3.1.1 OMI

In this study, the GEMS cloud retrieval algorithm was designed and developed based on the OMI cloud retrieval algorithm, which is based on $O_2$-$O_2$ absorption. OMI data were used as testing input to validate the algorithm. OMI, a UV-Vis observation satellite launched in 2004 aboard Aura, a satellite in the A-train, monitors the 270–500 nm band at intervals of 0.6 nm (Levelt

et al., 2006). In addition to ozone, it also produces information on sulfur dioxide, nitrogen dioxide, and two types of cloud products: OMCLDRRS, which uses RRS near 388 nm (Joiner et al., 2006; Vasilkov et al., 2008), and OMCLDO2, which uses $O_2$-$O_2$ absorption between 460 and 490 nm (Acarreta et al., 2004; Stammes et al., 2008; Veefkind et al., 2016; Vailkov et al., 2018). The cloud pressure is determined by the scattering and absorption characteristics that vary according to the presence of clouds in OMCLDRRS and OMCLDO2. RRS determines cloud pressure based on a phenomenon in which absorption lines

formed by radiation emitted from the sun are filled by non-elastic scattering as they pass through the Earth's atmosphere. In contrast, OMCLDO2 determines cloud pressure by using the absorption characteristics generated by collisions of $O_2$ molecules at an almost constant mixing ratio in dry air. The RRS method produces pressure closer to cloud-top pressures compared with

the $O_2$-$O_2$ method (Sneep et al., 2008; Joiner et al., 2012). Additionally, the two algorithms are effective at detecting high and low clouds based on the scattering and absorption used in each algorithm.

OMI data can be used to evaluate the GEMS cloud retrieval algorithm by excluding errors caused by the satellite's orbit, viewing geometry, and resolution (Park et al., 2020; Park et al., 2021). We used OMI values for RAD and IRR on randomly selected days each month in 2007 as input data for the algorithm. OMI provides surface reflectivity values for 36 channels, which are essential input data for cloud retrieval, and the GEMS cloud retrieval algorithm uses surface reflectivity values at 463 nm (adjacent to the $O_2$-$O_2$ absorption band), as does the OMI algorithm (Kleipool et al., 2008).

### 3.1.2 TROPOMI

The TROPOMI uses $O_2$-A absorption around 760 nm to produce cloud products for cloud correction (Loyola et al., 2018). The products include cloud fraction (CF), cloud-top and cloud-bottom pressure (CTP and CBP), cloud optical thickness, and various other cloud-related information. Launched in 2017 as a successor to environmental observation satellites, TROPOMI is equipped with four spectrometers that observe from the UV to shortwave-infrared wavelengths, including 760 nm, and can

provide greater accuracy in surface and cloud products compared with GEMS. TROPOMI is in SSO (as is OMI) and has a spatial resolution of $3.5 \times 5.5$ km, similar to that of GEMS (Latsch et al., 2022), making it useful for testing and validating the initial GEMS results. Therefore, we verified the retrieved cloud products using TROPOMI data.

TROPOMI observes the UV-Vis spectrum from its SSO and is primarily used to provide experimental and reference data for GEMS algorithms (Wang et al., 2020; Kang et al., 2020; Choi et al., 2020; Lee et al., 2020; Baek et al., 2022). TROPOMI data

were also used as input for cloud products in evaluating the performance of GEMS cloud retrieval algorithms. In this study, TROPOMI provided RAD, IRR, surface reflectance, pressure, and observation geometry for the evaluation of cloud algorithm performances. In addition, to analyze the characteristics of clouds observed from both a GEO and an SSO, verification was performed using data observed on the same day and time (within 15 minutes) as the GEMS observations.

### 3.1.3 CALIOP

The CALIOP satellite, which is often used for qualitative cloud verification, produces a vertical feature mask (VFM) of clouds using active sensors. CALIOP is a lidar instrument that was launched in 2006 as part of the Cloud-Aerosol Lidar and Infrared Pathfinder Satellite Observation (CALIPSO) program, which observes clouds and aerosols with a significant impact on Earth's weather and climate (Winker et al., 2009). It receives backscattered energy in bands at 532 nm and 1064 nm using three receivers and provides information on the vertical distribution of clouds and aerosols. In this study, we conducted a qualitative

comparison of the cloud products generated by different bands using L2-VFM data that represents a cloud-aerosol vertical existence.

### 3.1.4 AMI

A comparison of the cloud-detection characteristics of meteorological and environmental satellites with different spectral ranges that operate in a GEO can provide insights into the differences in cloud properties depending on the spectral band. AMI, which produces the most diverse cloud products, including pressure, phase, optical thickness, and effective particle radius, is a meteorological sensor onboard GK2A, which has 16 bands between 0.5 and 13.3 μm. Since the main purpose of weather observation is to gather data on weather phenomena, multiple cloud properties are defined and produced, and cloud detection is performed simultaneously using the visible and infrared regions. CTP is based primarily on infrared channels (Kim et al., 2019), and cloud pressure in this case is less sensitive to low clouds due to reduced thermal contrast (Miller et al., 2012).

### 3.2 Validation Method

Validation was conducted in two ways: by evaluating the performance of the algorithm itself, excluding differences in observation sensors; and by understanding the characteristics of the cloud products produced from actual observations in GEMS. To evaluate algorithm performance, radiance data and cloud products from OMI and TROPOMI, which make observations in the UV-Vis range, were used. To compare the GEMS cloud products, TROPOMI, CALIOP, and AMI, which collect data at a time similar to that of GEMS, were used as validation instruments (Sneep et al., 2008; Kim et al., 2019).

To evaluate the performance of the GEMS cloud retrieval algorithm using OMI and TROPOMI spectral ranges, no collocation process was required. In addition, for OMI, which is based on the same cloud model as GEMS, the cloud definitions derived were identical in terms of ECF and CCP, making it possible to avoid preprocessing. However, for TROPOMI, although clouds were assumed to be LER, as was the case with GEMS, the cloud albedo (CA) was retrieved, not 0.8, and the TROPOM CF had to be converted to cloud effective amount prior to making comparisons using Eq. (4) and the ECF of GEMS (Loyola et al., 2018, Latsch et al., 2022).

$$ECF = CF \times \frac{CA}{0.8} \tag{4}$$

Moreover, the cloud pressure from TROPOMI was provided as separate cloud-top and cloud-base pressures and cloud pressure (CP) with MLER assumption with retrieved CA. To compare its data with those from GEMS, which only provides cloud centroid pressure, the CP was used for validation.

To verify various cloud characteristics, cases of high aerosol concentration, typhoons, and fog were selected when GEMS, AMI, TROPOMI, and CALIOP satellites passed over the Korean Peninsula. To validate satellite data from different orbits, spatiotemporal collocation was necessary. TROPOMI and CALIOP passed the GEMS field of view between 03 and 06 UTC, we use the entire swath to validate the GEMS products, but since most cases occurred close to the Korean Peninsula, we primarily used the 04 UTC observation for validation. GEMS observes the Korean Peninsula at 45 minutes past every hour during daytime, therefore, data observed at 0445 UTC was selected as the validation data. AMI observed the Korean Peninsula every 10 minutes, data observed at 0440 UTC was used for validation.

AMI provides cloud-detection results at a spatial resolution of 2 km × 2 km. To compare them with the ECF and to select pixels included in the GEMS observations, collocation was performed and the CF was calculated by counting the number of cloud pixels for comparison. CTP is the mean value of the cloud pixels. However, quantitative verification of the CCP from GEMS and CTP from CALIOP or AMI can be difficult, as they represent cloud pressures with different definitions. A VFM can evaluate the validity of cloud detection and estimates as it provides cloud vertical distribution. The CCP is converted to cloud centroid height (CCH) using Eq. (5) for qualitative verification of cloud pressure:

$$\text{CCH} = -h_{scale} \ln \left( \frac{CCP}{p_s} \right) + h_s \tag{5}$$

where $p_s$ is the surface pressure and $h_s$ is the surface altitude of GEMS pixel. A value of 7,710 m was obtained by fitting the results to the atmosphere profile from the DISCOVER-AQ campaign (Flynn et al., 2014), which is a reference in the GEMS algorithm. For spatial collocation, the nearest neighbor method was used based on based on CALIOP observation pixels, which have the narrowest swath. The same method was applied to GEMS and TROPOMI collocation, along with a comparison of the nearest pixels.

## 4 GEMS Cloud Retrieval Algorithm Results Using OMI and TROPOMI Radiances

### 4.1 Comparison with OMI Clouds

The performance of the GEMS cloud retrieval algorithm was evaluated using the theoretical background of the OMI's OMCLDO2 algorithm. Using the same input data is valuable for understanding algorithm differences without errors from the sensor. Since OMI is the foundation of the GEMS cloud retrieval algorithm, the GEMS cloud retrieval algorithm would produce a product highly correlated with the OMI cloud product. Fig. 2 displays the characteristics of cloud products based on seasonal observational data for random days of each month in 2007. To represent the seasons, the analysis was performed using observational data for December, January, and February (DJF) for winter, March, April, and May (MAM) for spring, June, July, and August (JJA) for summer, and September, October, and November (SON) for autumn. For comparing the two products, we presented the density scatterplots with their correlation analyzed through linear regression.

The ECF values from GEMS and OMI were similar across the entire study area, regardless of cloud amount. The correlation coefficient was high, ranging from about 0.92 to 0.95, and the slope of the regression line was also close to 1, ranging from 0.9 to 0.94, indicating a strong agreement between GEMS and OMI. The root mean square error and mean bias error were 0.12 and −0.01, respectively, with notable errors primarily in areas with small ECFs. Overall, the correlation coefficient and slope for the ECF between the two satellites did not show significant seasonal differences, suggesting that the algorithm design did not result in significant seasonal bias.

As with the case of the ECF, GEMS produced CCP values similar to those obtained from OMI across the entire domain. However, many previous studies (Vasilkov et al., 2008; Sneep et al., 2008; Loyola et al., 2018; Compernolle et al., 2021)

reported that the accuracy of CCP retrieval using $O_2$-$O_2$ absorption significantly decreases in areas with an ECF less than 0.2. Only pixels with an ECF greater than 0.2 were validated. The correlation coefficient was approximately 0.95 for all seasons,

and the slope of the regression line was greater than 0.9, indicating that GEMS produced a CCP similar to those from OMI regardless of season.

Fig. 3 displays the outcome of the GEMS cloud retrieval process, derived from OMI observations conducted on March 25, 2007. We selected three swaths for 0300 to 0600 UTC for the OMI data, covering the GEMS field of view. Fig 3(a) and (b) show the ECF retrieved by OMI and GEMS, respectively. Fig. 3(c) shows the difference between the ECF retrieved by OMI

and GEMS. Fig. 3(d) and (e) show the CCP retrieved by OMI and GEMS, respectively, while Fig. 3(f) shows the difference between the CCP retrieved by the two algorithms. The retrieved ECF had similar errors in each swath, with the errors being most prominent in areas with a low ECF and the edge in RAA. These errors were strongly correlated (approximately 0.8) with RAA. Given GEMS's geostationary orbit, characterized by a reduced discontinuity in azimuth angle compared to polar-orbiting satellites like OMI, consideration of RAA was minimized in the algorithm. This could be a major source of the

differences between the two algorithms.

## 4.2 Comparison with TROPOMI Clouds

We evaluated the performance of the GEMS cloud retrieval algorithm using TROPOMI data, collected simultaneously with GEMS data, considering orbital differences and the spectral ranges of observation. Similar to the experiments in Section 4.1, we conducted both quantitative validation (Fig. 4) and qualitative validation (Fig. 5) for randomly selected days in March,

June, September, and December 2021. Since TROPOMI cloud retrieval products are defined differently from GEMS products, we converted the CF provided by TROPOMI to an ECF using Eq. (4) to represent the cloud amount. We then applied the CP for validation with the CCP. The cloud retrieval and analysis for three swaths (observation time 0300–0600 UTC) covering East Asia within the GEMS field of view. Fig. 4 shows the results for March, June, September, and December from left to right, with the top panels showing the results for the ECF and the bottom panels showing the density scatterplot for the CCP.

There was no noticeable difference in the algorithm's performance across seasons. However, for the ECF, due to the correction and comparison products with different definitions, we found that the linear relationship was not clear. The correlation coefficients were relatively low compared with the values in Fig. 2.

In Fig. 5, the ECF (top) and CCP (bottom) obtained from TROPOMI (left) and GEMS (right) on September 16, 2021, are presented. The difference in the ECF is pronounced in the Tibetan Plateau, and overall, there is a noticeable difference in the

ECF between ocean and land areas. This is due to the fact that TROPOMI does not provide surface reflectance at 463 nm and instead uses surface reflectance at 420 nm as input data. As for the CCP, a direct comparison with the TROPOMI CP is challenging due to different definitions, but overall, the GEMS CCP was observed at a lower altitude than the TROPOMI CP.

**5 GEMS Cloud Retrieval Algorithm Results Using GEMS Radiances**

The accuracy and usability of the cloud products retrieved by the GEMS cloud retrieval algorithm were evaluated by comparing

them with products produced by other sensors and orbits, such as TROPOMI, AMI, and CALIOP, using data collected after an in-orbit test in 2021. Specifically, cases of high concentrations of air pollutants, typhoons, and sea fog were selected to reflect the atmospheric environment and cloud characteristics over East Asia.

**5.1 Air Pollution Case**

As the main purpose of GEMS is to monitor air pollutants in East Asia, we selected a high-air-pollutant case. We analyzed the

characteristics of clouds that appear during high concentrations of air pollutants, as was the case in May 2021, Fig. 6 and 7 show cloud detection and cloud pressure, respectively. The data analyzed were taken at approximately 0430 UTC, coinciding with the passage of A-train satellite over the region. Subsequently, the 0445 UTC data were analyzed based on the GEMS field of view. Fig. 6(a) shows the ECF result from GEMS, while Fig. 6(b), 6(c), and 6(d) shows the ECF result from TROPOMI, the CF image of AMI, and an image of the aerosol optical depth (AOD) derived from AMI, respectively.

Comparing GEMS and AMI clouds, it is clear that AMI identifies many more areas as clouds compared with GEMS as it was designed to observe meteorological phenomena and is particularly sensitive to clouds. The cloud fractions from TROPOMI and GEMS in Fig. 6(c) show similar pattern, confirming that both instruments detect clouds at a similar degree of sensitivity. The GEMS cloud retrieval algorithm uses surface reflectance climatology from OMI as input, allowing the exposure of the surface properties in the ECF, as the surface reflectance data is not updated. In Fig. 6(d), which represents the AMI AOD, a

high-concentration air pollutants in the southern region of the Korean Peninsula are clearly visible. Both GEMS and TROPOMI detected a thin cloud (ECF < 0.2) in this region. As the cloud-production algorithms of GEMS and TROPOMI do not provide additional information on aerosols, it was difficult to classify aerosols from cloud data, highlighting a limitation of the GEMS cloud retrieval algorithm.

Fig. 7 shows the CCP validation results for the same day as Fig. 6. Fig. 7(a) displays the GEMS CCP, while Fig. 7(b) shows

the TROPOMI mCP. Fig. 7(c) shows the CTP provided by AMI, and Fig. 7(d) is an image overlaying the cloud products of three satellites on the CALIOP VFM. To compare with CALIOP VFM, the CCP analysis figures, such as Fig. 7, use height units in Eq. (4).

The cloud height results displayed in Fig. 7(a)–(c) exhibit distinctive characteristics for each satellite, making it difficult to find commonalities. GEMS values were similar to surface height for clear pixels, while for areas with aerosols, the retrieved

CH values were within the range associated with aerosols. The algorithm used by GEMS cannot distinguish between clouds and aerosols. The scattering effect at the wavelengths used for cloud retrieval causes the cloud altitude to be calculated as if there were clouds, rather than the aerosols that are actually present due to the reflection effect caused by aerosols. Comparing TROPOMI CH with GEMS CH revealed that, in general, GEMS tends to agree with TROPOMI CH in low cloud (less than 6 km), but estimates lower cloud heights than TROPOMI cloud pressure calculations for high cloud (over 6 km). The scale

pressure of $O_2$-$O_2$ absorption is approximately 700 hPa, and this altitude becomes the reference for the relationship between TROPOMI and GEMS cloud pressure. Consequently, while lower clouds display cloud heights comparable to TROPOMI, clouds at higher altitudes have the characteristic of estimating lower altitudes.

Due to the GEO orbit of GEMS, there are temporal variations in SZA as well as several input variables. Using the time-specific (04:45 UTC) observation data of the air pollution case, we conducted testes to examine the error when input variables over

time were inaccurately entered as fixed values. The results indicate that the largest error in cloud calculation occurs when the change in SZA is not adequately accounted for. The surface reflection, which causes an approximately 1% change in time, also caused a significant error, with substantial differences observed in land-ocean characteristics. On the other hand, the error caused by a minor change in ground pressure over time was insignificant.

## 5.2 Typhoon Case

Typhoons are meteorological phenomena that involve wide variety of clouds, making them a useful case for cloud verification. Typhoon Chantu, which occurred in September 2021, was a large typhoon to the southwest of the Korean Peninsula, as shown in Fig. 8(a)–(c). Fig. 8 also contains the cloud detection results of each satellite, as in Fig. 6, but excludes verification of AOD with a focus on clouds. Given the well-developed tropical low-pressure system, typhoons are accompanied by various types of clouds, ranging from thick clouds to cirrus formations. This provides valuable insight into how GEMS retrieves different

cloud types. In Fig. 8(a), significantly high ECF is observed around the eye of the typhoon. TROPOMI and AMI also shows high agreement in cloud detection, particularly for thick clouds.

The clouds that come with typhoons are useful for understanding the general characteristics of cloud distribution. However, in cases where clouds are sufficiently thick, the active sensor on CALIOP has limited penetration and may not provide enough information about the underlying cloud structures. Comparing cloud height from Fig. 9(a) to (c), GEMS cloud height is lower

than the heights estimated by other satellites in general. Therefore, we analyzed the vertical structure of the clouds for details in Fig. 9(d), where the VFM of typhoon clouds reports on typhoon clouds exclusively in the upper atmosphere due to their thickness. AMI, which produced the cloud-top height in Fig. 9(d), indicates cloud existence at high altitudes consistent with the CALIOP VFM, except for the typhoon eye. Both GEMS and TROPOMI accurately detected the typhoon eye, and, as shown in Fig. 8, GEMS tended to underestimate cloud height compared with TROPOMI in high and thick cloud conditions.

In addition, while AMI produced CTH in the upper atmosphere for areas estimated to be multi-layered above 40°N latitude, the GEMS CCP tended to produce cloud height in the lower layer. These characteristics can be useful for understanding the properties of multi-layered clouds in further studies.

Despite the infeasibility of verifying all GEMS observation times with other satellites, analysis of cloud production characteristics over time is required. In the case of typhoons in which clouds are transparent and move rapidly over time, a

qualitative analysis of cloud movements over time was conducted. It was discovered that GEMS cloud retrieval algorithm accurately identified the movement of typhoons over time.

**5.3 Sea Fog Case**

Sea fog is a common phenomenon on the western coast of Korea and can be included in the transport path of air pollutants. False detection of sea fog can lead to significant errors in observations of atmospheric pollutants using GEMS. In Fig. 10 and Fig. 11, sea fog is difficult to distinguish from low-level clouds over the ocean using only the naked eye, but it can be identified by observing its movement over time from a geostationary orbit. Unlike clouds, sea fog tends to linger near the coast and has little flow. All three cloud fractions, GEMS, TROPOMI, and AMI, mistook the presence of sea fog as evidence of clouds.

Due to sea fog's high reflectance, accurate retrieval of CCP is essential for correcting atmospheric pollutant observations. In the case of GEMS, sea level pressure can be obtained for areas suspected of being susceptible to sea fog, and it is useful for correcting the influence of sea fog on atmospheric component retrievals. TROPOMI, which uses the near-infrared region for cloud retrieval, also retrieves CP in sea fog–prone areas close to the surface altitude. AMI shows a tendency to retrieve cloud heights similar to those of low-level clouds. If the cloud height in GEMS is overestimated, as in the case of AMI, it may overestimate atmospheric components.

**5.4 Monthly Cloud Product Validation**

For the monthly validation of GEMS cloud products, we randomly selected 2 days of each month from 2021 to 2022 and conducted the validation against TROPOMI data. To exclude the influence of variations in GEMS observation areas due to changing seasons, we employed the full west mode and selected the times when TROPOMI observation paths were present for the validation. Collocation was performed using the same method as described in Section 4.2, to assess the cloud products from both satellites. For certain periods, TROPOMI provides cloud products in both the OFFL (Offline) and RPRO (Reprocessed) versions, so we presented the correlation coefficients from the validation using both products (Fig. 12). In addition, we accounted for land cover since the precision of cloud product retrievals can vary between ocean and land due to factors like surface reflectance, as indicated by the results of scene analyses.

For 2 years of monthly validation results, in the case of ECF, there appeared to be no significant monthly variations in accuracy. Generally, higher accuracy was observed over ocean compared to land. Furthermore, the difference in validation results based on TROPOMI versions was not prominent. On the other hand, for CCP, substantial monthly variations in accuracy were observed, especially a noticeable decrease in CCP correlation coefficients during the summer seasons (June, July, and August) over ocean. Additionally, variations in accuracy were evident depending on TROPOMI versions, with the newly provided RPRO version showing improved correlation with GEMS.

The difference in ECF accuracy based on land cover can largely be attributed to the use of OMI climatology values for surface reflectance as input data. It is expected that this accuracy difference between the land and ocean based on land cover will significantly decrease when surface reflectance data observed by GEMS is applied as inputs.

## 6 Conclusions and Discussions

In this study, we introduced, compared, and validated the GEMS cloud retrieval algorithm and its products. The GEMS cloud retrieval algorithm is based on an algorithm developed for the polar-orbiting OMI satellite, and produces values similar to

OMI's cloud products, with a correlation coefficient greater than 0.94 for all seasons. Although it was difficult to make quantitative comparisons due to different definitions of clouds, we also validated GEMS data using data from TROPOMI, a UV-Vis environmental satellite currently in operation. The ECF was comparable to TROPOMI in general. Also, compared to TROPOMI CP, the CCP was overestimated at pressures lower than scale pressure and underestimated at pressures over than scale pressure. It demonstrates its suitability for use in trace-gas corrections.

As GEMS operates in a geostationary orbit, unlike previous environmental satellites, it is important to detect diurnal cloud characteristics over East Asia. In this study, cases of air pollutants, typhoons, and sea fog, all of which are common in East Asia, were selected to compare and verify the GEMS cloud data with those of other satellites. Cloud heights produced by GEMS, TROPOMI, and AMI were compared based on a CALIOP VFM, which can observe the vertical distribution of clouds. CALIOP was found to be insufficient for detecting thick, low-altitude clouds. The GEMS cloud retrieval algorithm is sensitive

to high pressure greater than scale pressure; as a result, GEMS estimates of cloud height were the lowest among the four satellites, corresponding to the height at which clouds reflect radiation. The cloud-top height produced by TROPOMI was located at a lower altitude compared with the AMI CTH, and GEMS CH usually followed TROPOMI CH. AMI generally produced cloud-top heights that closely matched the top of CALIOP VFM, but tended to overestimate cloud height in the case of sea fog. Through comparisons of GEMS ECF and CCP, we confirmed that the cloud characteristics were well reflected in

the retrieval results, making it useful for correcting the beam path length of observed radiation.

The current version of the GEMS cloud retrieval algorithm generates results consistent with the output produced by other satellites and accurately reflect cloud characteristics in East Asia. However, two issues have been pointed out. First, the reflectivity and pressure characteristics of inland regions, including the Tibetan Plateau, are directly reflected in the derived ECF, and (2) the ECF was greater than zero in most regions. This is due to the current algorithm using OMI climatological

data as input values for surface reflectance. The surface reflectance from OMI is difficult to represent due to the influence of changes in background aerosols in the atmosphere, among other factors (M. Kim et al., 2021). In the next version of the algorithm update, we plan to replace the input value with surface reflectance observed by GEMS, which is expected to address both problems. Since the GEMS cloud is retrieved through the DOAS method using linear fitting, it is necessary to compare the results with nonlinear fitting. One approach for applying nonlinear fitting is to utilize QDOAS, as demonstrated by

Danckaert et al. (2012). We plan to analyze the results of cloud retrievals using the nonlinear fitting method in existing algorithms in future research. Also, QDOAS application enables the consideration of $NO_2$ absorption. We anticipate the improvement of the GEMS cloud retrieval algorithm in the future through considering these remaining issues.

## Financial support

This subject is supported by Korea Ministry of Environment (MOE) as Public Technology Program based on Environmental Policy (2017000160003) and was supported by a grant from the Nation Institute of Environmental Research (NIER), funded by the Korea Ministry of Environment (MOE) of the Republic of Korea (NIER-2023-04-02-050).

## Data availability

OMI, TROPOMI, and CALIPSO products can be downloaded from https://search.earthdata.nasa.gov/search. GK2A product can be downloaded from https://datasvc.nmsc.kma.go.kr/datasvc/html/main/main.do. The results of this study were conducted with GEMS CLD Version 3 from the Nation Institute of Environmental Research (NIER), which will be released near future from https://nesc.nier.go.kr/ko/html/index.do.

## Author contributions

BRK, YSC, and JK contributed to the study conception and design. Material preparation, data collection, analysis and visualization were performed by GK and MC. The first draft of the manuscript was written by BRK. Critical comments and revisions were provided by YSC and JK. All authors read and approved the final manuscript.

## Competing interests

At least one of the (co-)authors is a member of the editorial board of *Atmospheric Measurement Techniques*. The peer-review process was guided by an independent editor, and the authors also have no other competing interests to declare.

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

Table 1: Values of each nodal point used to calculate the look-up table in the GEMS cloud retrieval algorithm.

| Parameters | Values |
|---|---|
| Solar zenith angle (°) | 0, 18.2, 25.8, 31.8, 36.9, 41.4, 45.6, 49.5, 53.1, 56.6, 60.0, 63.3, 66.4, 69.5, 72.5, 75.5, 78.5, 81.4, 84.3 |
| Viewing zenith angle (°) | 0, 18.2, 25.8, 31.8, 36.9, 41.4, 45.6, 49.5, 53.1, 56.6, 60.0, 63.3, 66.4, 69.5, 72.5, 75.5, 78.5, 81.4, 84.3 |
| Relative azimuth angle (°) | 0, 90, 180 |
| Surface pressure (hPa) | 1014, 1000, 950, 900, 850, 800, 750, 700, 600, 500, 400, 300, 200, 100 |
| Surface reflectance | 0, 0.05, 0.1, 0.2, 0.3, 0.4, 0.5, 0.6, 0.7, 0.8, 0.9, 1.0 |
| Effective cloud pressure | 0, 0.05, 0.1, 0.15, 0.2, 0.25, 0.3, 0.35, 0.4, 0.45, 0.5, 0.55, 0.6, 0.65, 0.7, 0.75, 0.8, 0.85, 0.9, 0.95, 1.0 |
| Cloud centroid pressure (hPa) | 1014, 1000, 950, 900, 850, 800, 750, 700, 600, 500, 400, 300, 200, 100 |


**Table 2: Overview of the cloud products included in this study.**

| Instrument | Spectral range (nm) | Cloud product name | Variable | Cloud spectral range (nm) |
|---|---|---|---|---|
| OMI | 270–314, 306–380, 350–500 | OMCLDO2 | Effective cloud fraction<br>Cloud height | 460–490 |
| TROPOMI | 270–495, 710–775, 2305–2385 | ROCINN CRB | Cloud fraction<br>Cloud albedo<br>Cloud pressure/height | 758–766 |
| GEMS | 300–500 | GEMS CLD | Effective cloud fraction<br>Cloud centroid pressure | 460–485 |
| AMI | 470, 511, 640, 856, 1380, 1610, 3830, 6241, 6952, 7344, 8592, 9625, 10403, 11212, 12364, 13310 | GK2A CTH | Cloud top height | 8592–13310 |
| CALIOP | 532, 1064 | VFM | Vertical feature mask | 532, 1064 |

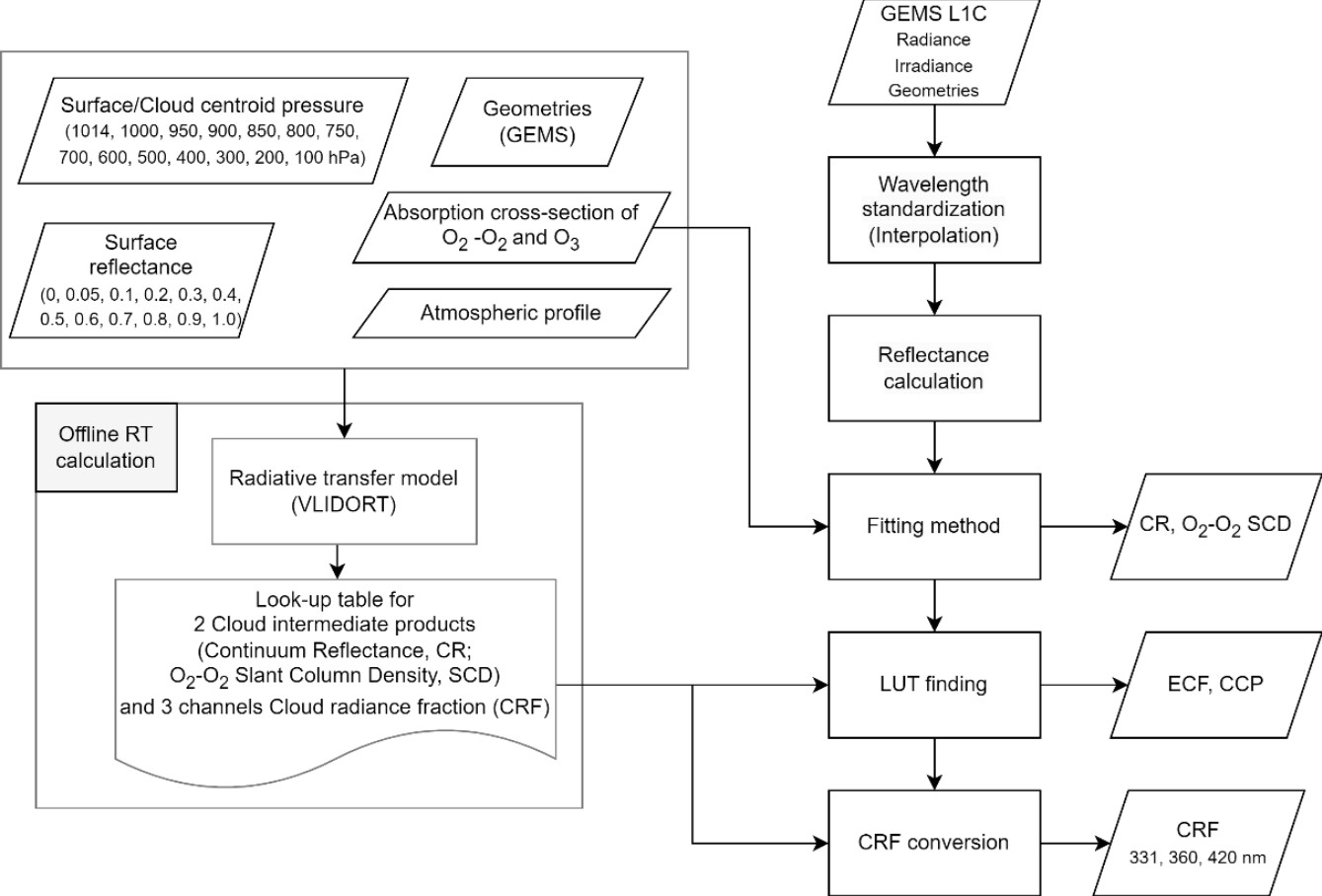

**Figure 1: GEMS cloud retrieval algorithm flowchart.**


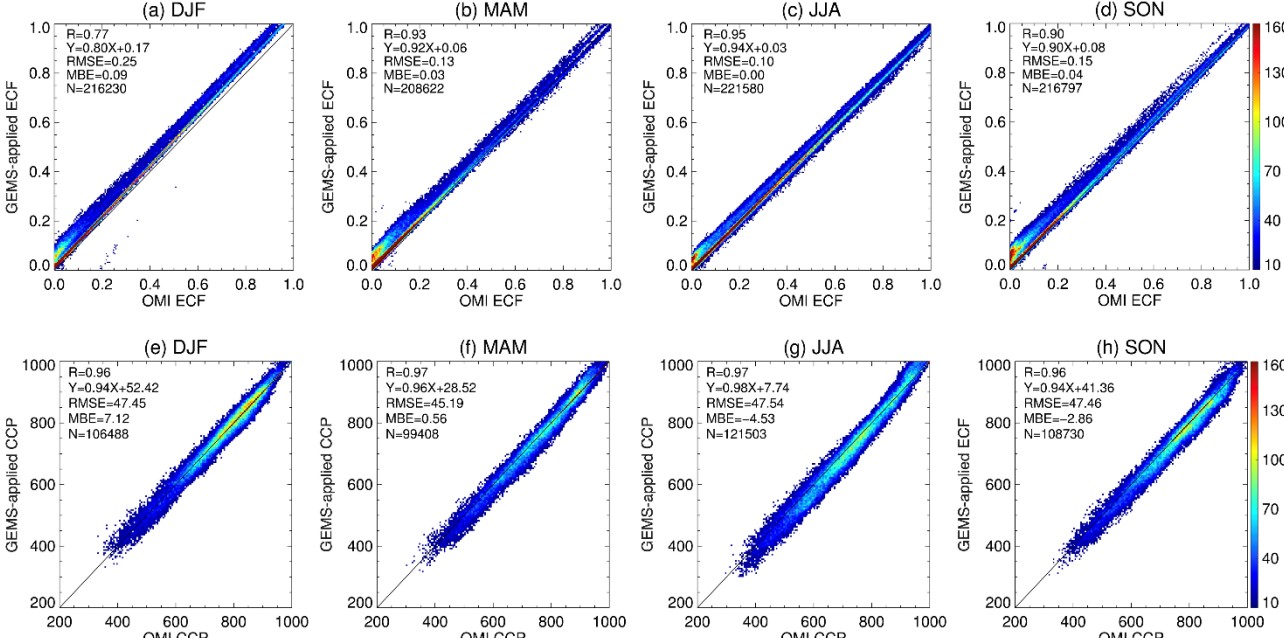

**Figure 2: Cloud products for random days in each month of 2007 (classified by seasons: DJF-winter, MAM-spring, JJA-summer, SON-autumn) for comparison with cloud products from the OMI algorithm. The top panel shows the density scatterplot for effective cloud fraction (ECF) and the bottom panel shows the density scatter plot for cloud centroid pressure (CCP). The x-axis represents the values from OMI, and the y-axis represents the cloud products from the GEMS algorithm. The solid line indicates the 1:1 line, and the correlation coefficient (R), regression equation, root mean square error (RMSE), mean bias error (MBE), and number of pixels (#) used in the analysis are presented.**


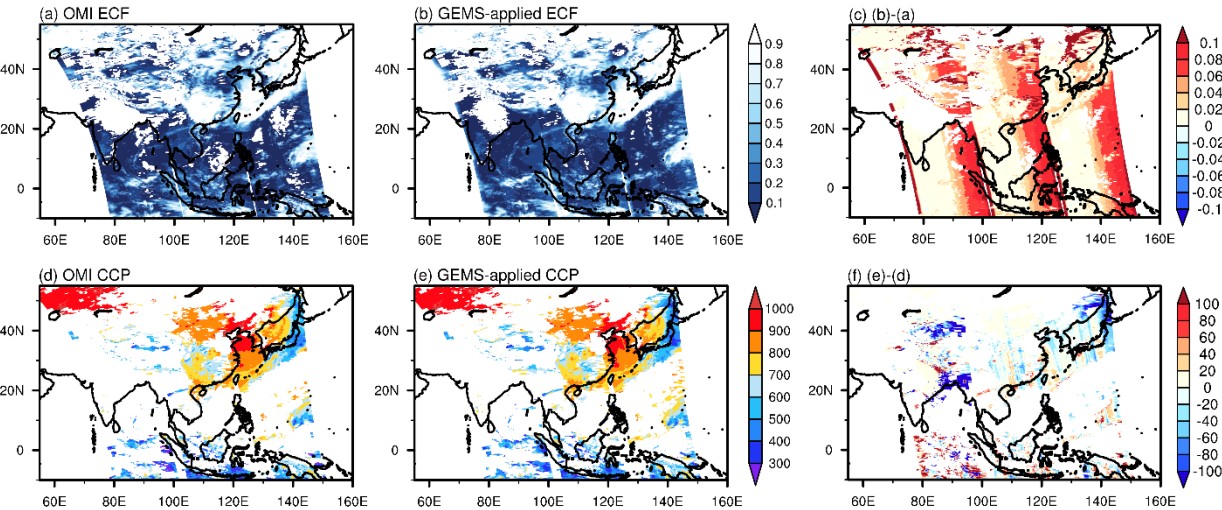

**Figure 3: Results of performance validation of the GEMS cloud retrieval algorithm using OMI data. The data from March 25, 2007, were used as input. (a) ECF provided by OMI. (b) ECF retrieved by GEMS using OMI L1B data. (d) CCP retrieved by OMI. (e) CCP image retrieved by GEMS using the OMI radiance data input same as (b). (c) and (f) Differences in cloud retrieval between the GEMS and OMI algorithms.**


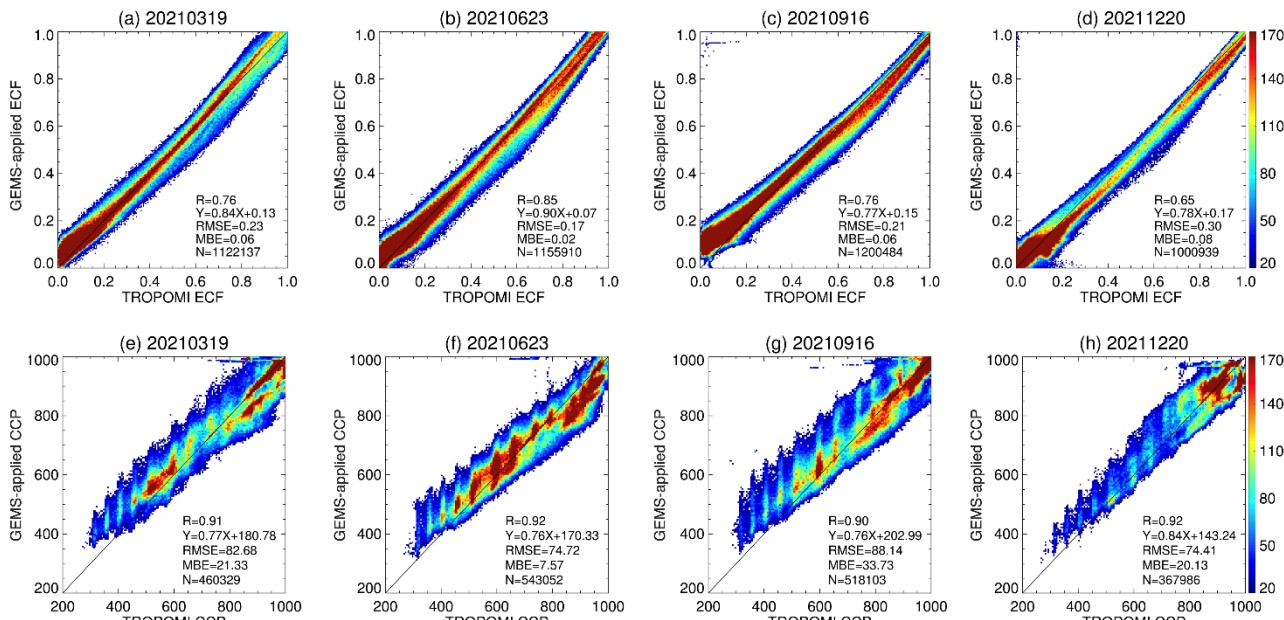

**Figure 4: This figure is similar to Figure 1 but for TROPOMI. To verify the performance and seasonal dependency of GEMS cloud retrieval algorithm, cloud retrieval results were compared with those of TROPOMI for random days in March, May, September, and December 2021. The top panel shows the results for ECF and the bottom panel shows the results for CCP in a density scatterplot. As definitions of cloud pressure differ between GEMS and TROPOMI, the density scatterplot was generated using the mean value of CTP and CBP (mean cloud pressure, mCP) from TROPOMI. The x-axis shows the TROPOMI values, and the y-axis shows the GEMS cloud retrieval results. The solid line represents the 1:1 line, and the correlation coefficient and regression equation are also displayed.**

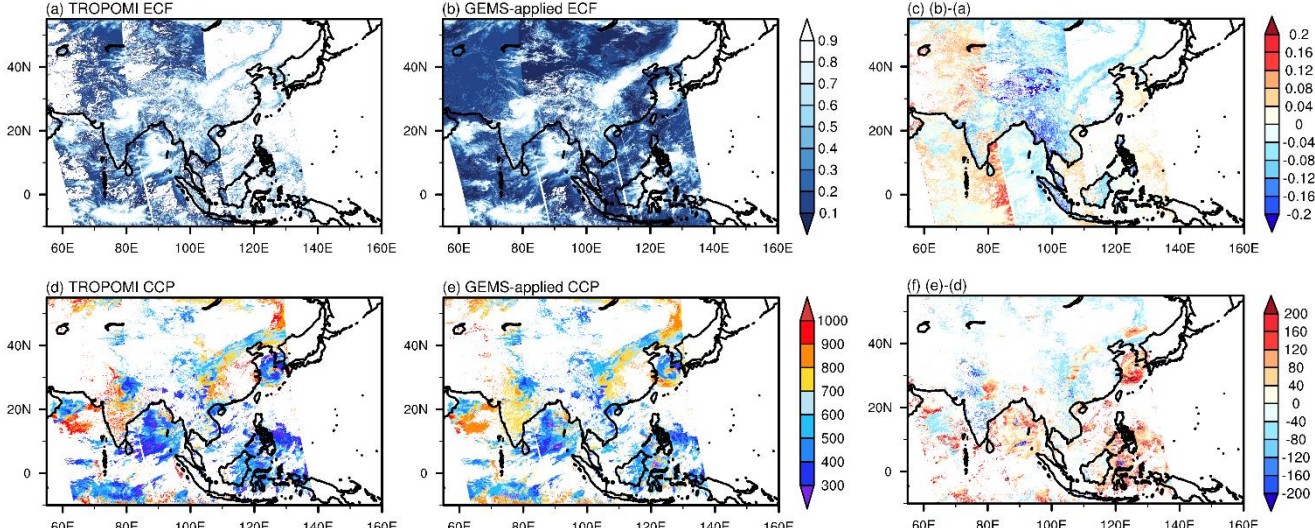

**Figure 5: This figure, similar to Figure 2, shows the validation of the GEMS cloud retrieval algorithm using TROPOMI data. Cloud retrievals were performed using observations from September 16, 2021, as input. (a) TROPOMI CF converted to ECF. (b) ECF derived from GEMS using TROPOMI L1B as input. (d) CP retrieved from TROPOMI. (e) CCP image retrieved by GEMS cloud retrieval algorithm. (c) and (f) Differences in cloud retrievals between GEMS and TROPOMI algorithms.**

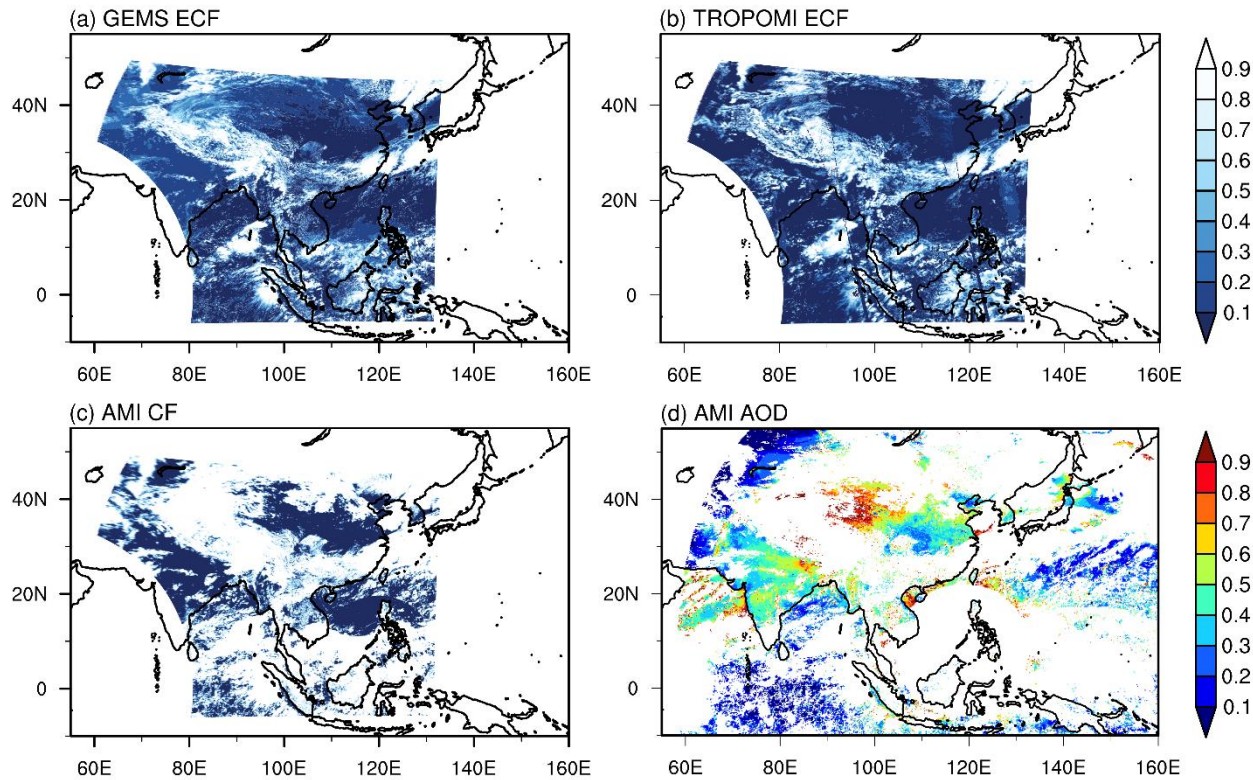

**Figure 6: Four different images of air pollutants as observed on May 8, 2021, in the GEMS field of view. (a) ECF retrieved by GEMS. (b) ECF calculated by converting the CF data from TROPOMI. (c) CF image calculated by collocating AMI data to GEMS pixels. (d) Aerosol optical depth (AOD) image obtained from AMI data.**

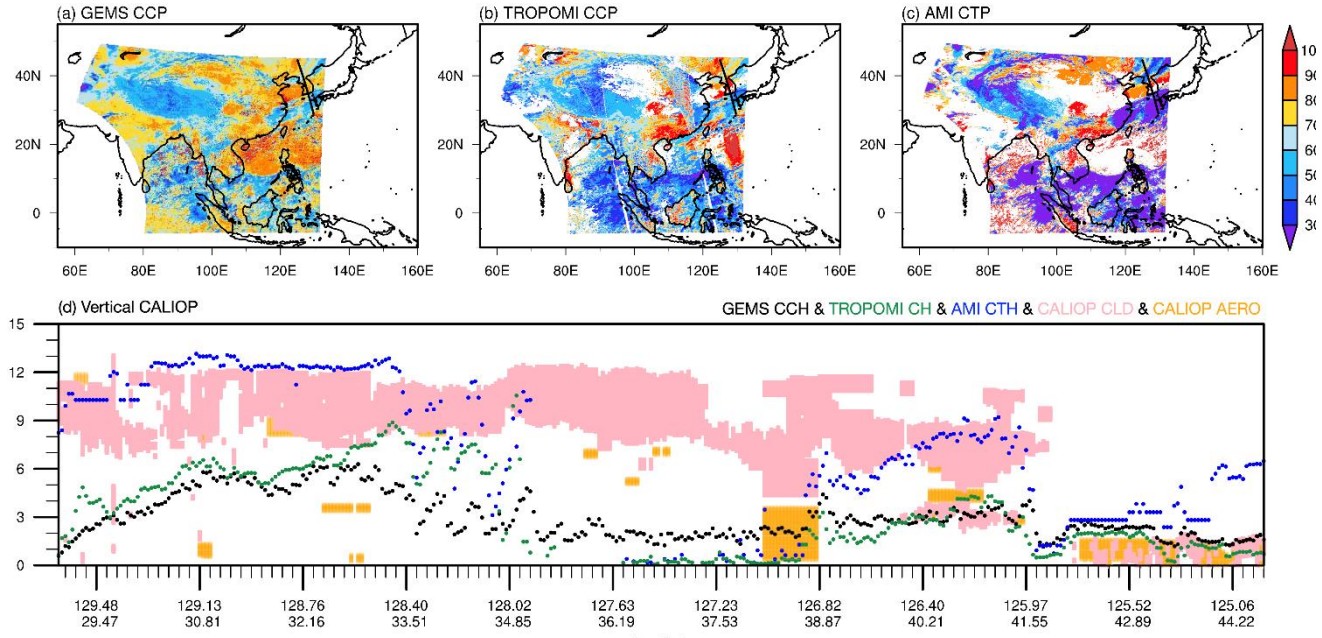

**Figure 7: Cloud pressure observations on May 8, 2021. (a) GEMS CCP. (b) TROPOMI mCP. (c) AMI CTP. (d) CALIOP VFM with cloud pressure products. In (a) to (c), the thick black line represents the scan path of CALIOP. In (d), a black dot represents GEMS CCP, a blue dot represents GK2A CTP, the gray shade represents between the TROPOMI CTP and CBP, the pink shade represents a CALIOP cloud mask, and the orange shade represents a CALIOP aerosol mask.**

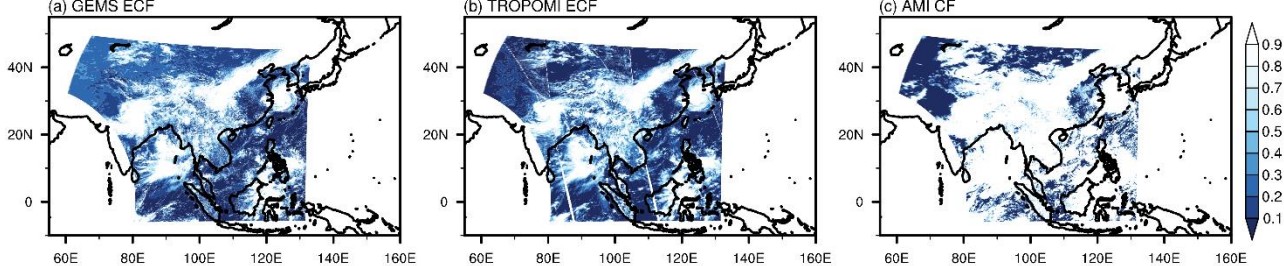

**Figure 8: Similar to Figure 5, an intercomparison of ECF is presented for the case of Typhoon Chantu, which occurred on September 16, 2021. (a) GEMS ECF. (b) TROPOMI ECF. (c) AMI CF.**

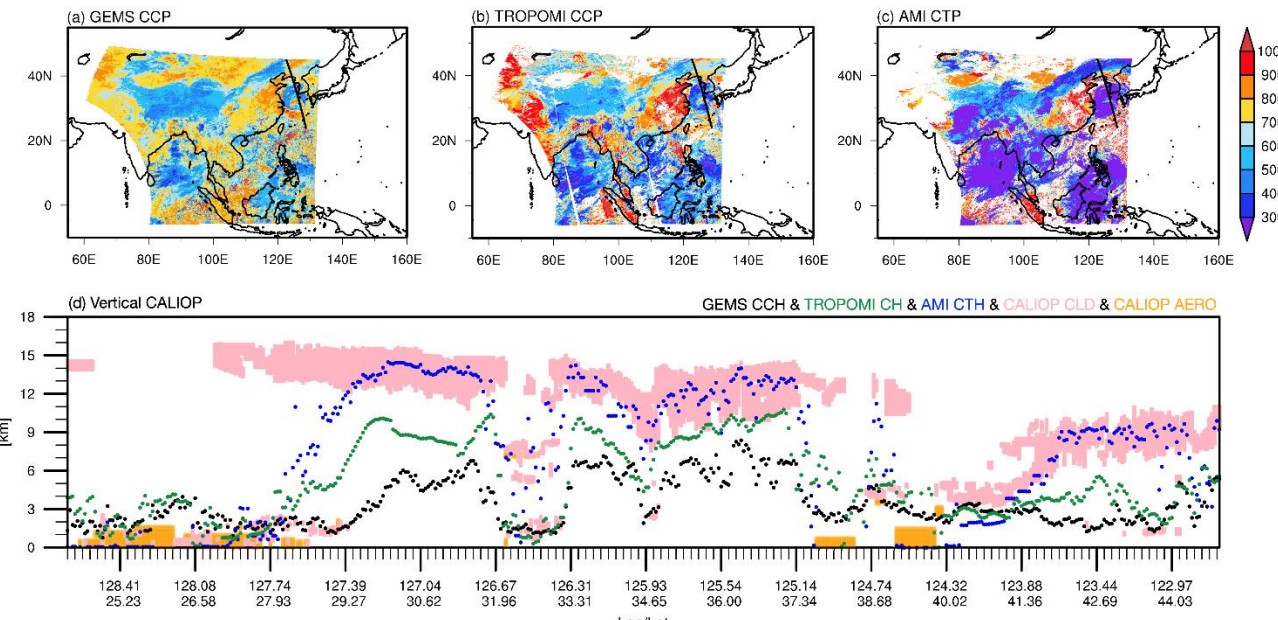

**Figure 9: The case of Typhoon Chantu on September 16, 2021. (a) CCP obtained from GEMS. (b) mCP from TROPOMI. (c) CTP derived from AMI. (d) CHs converted to cloud altitude from three satellites overlaid on the CALIOP VFM image.**

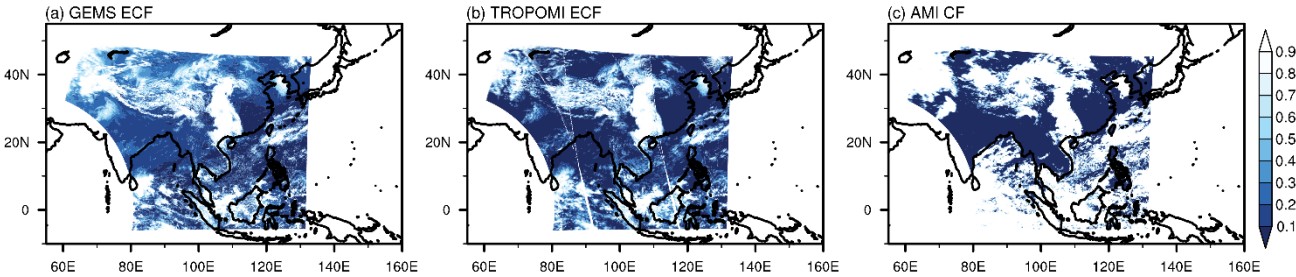

**Figure 10: Similar to Figure 7, intercomparison of ECF for the case of sea fog, which occurred on March 25, 2021. (a) GEMS ECF. (b) TROPOMI ECF. (c) AMI CF.**

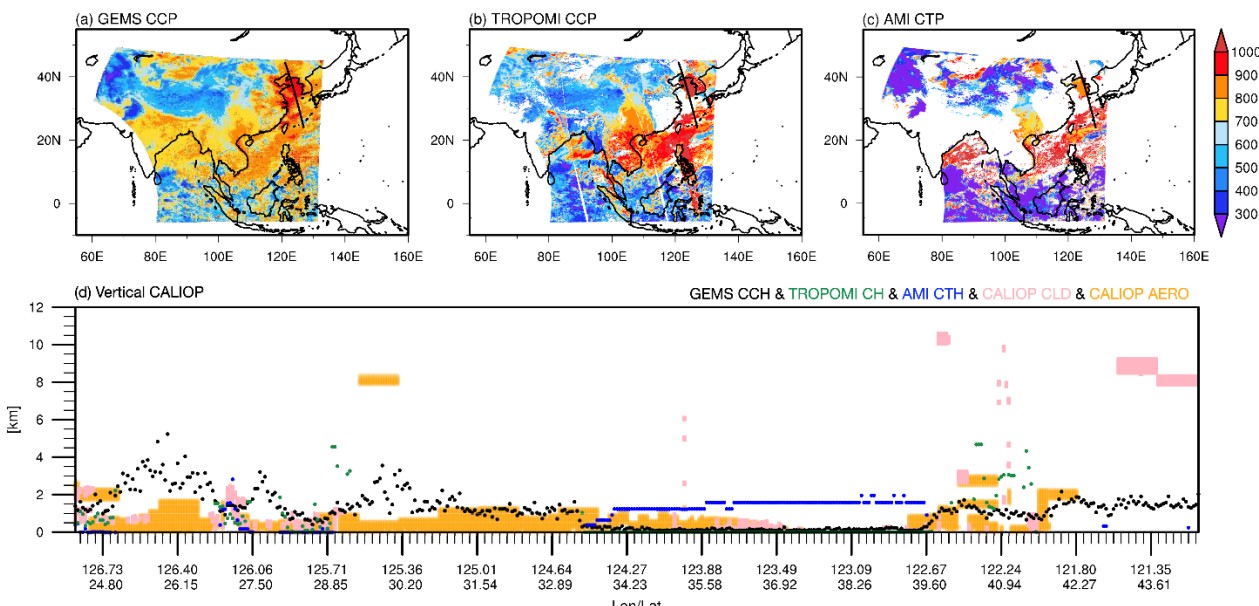

630

**Figure 11: Sea fog on March 25, 2021. (a) CCP obtained from GEMS. (b) mCP from TROPOMI. (c) CTP derived from AMI. (d) CHs converted to cloud altitude from three satellites overlaid onto the CALIOP VFM image.**

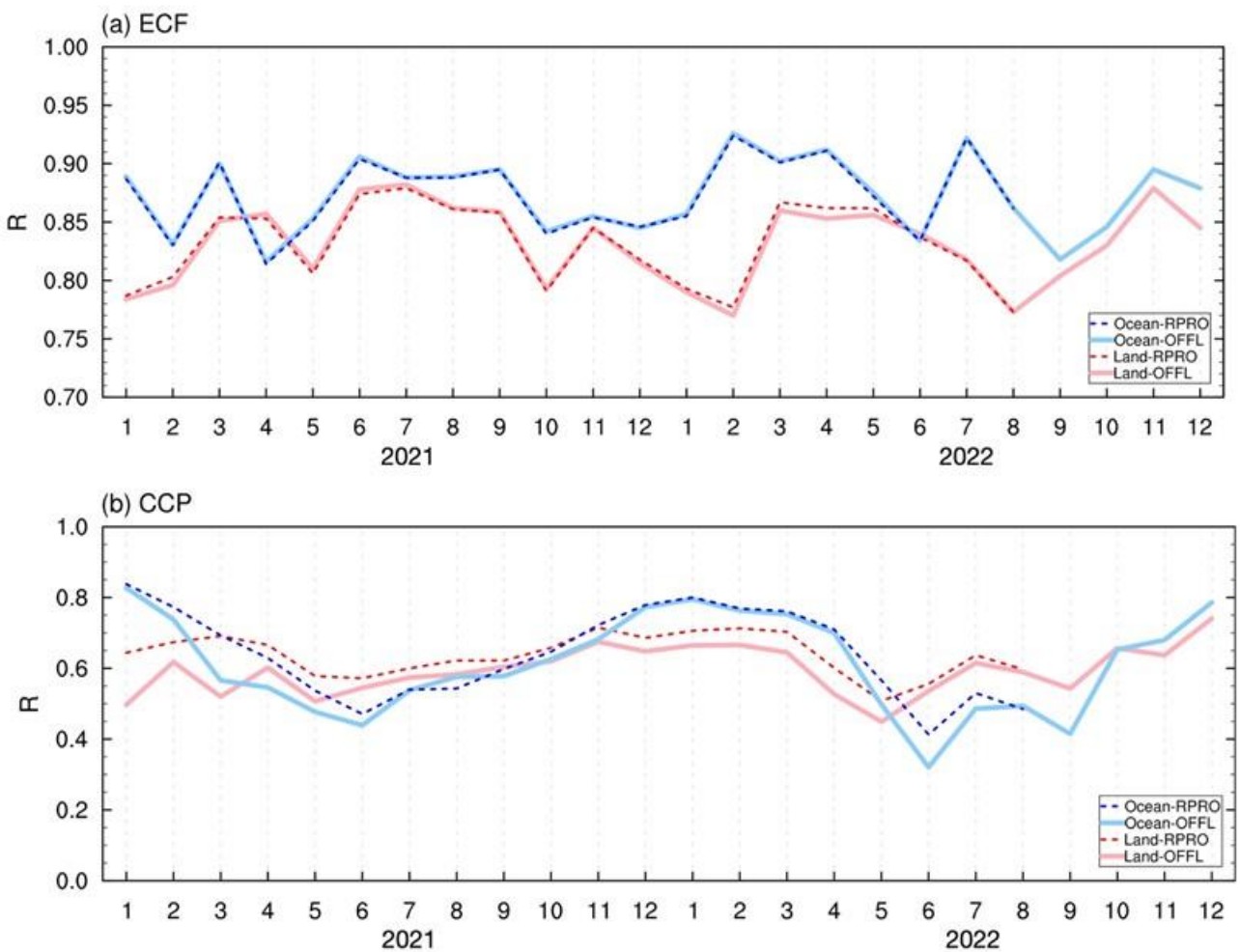

**Figure 12: The monthly correlation coefficient (R) values between GEMS cloud products and TROPOMI OFFL version (solid line) and between GEMS cloud products and TROPOMI RPRO version (dotted line) are presented in Figures (a) for ECF and (b) for CCP. The red and blue color respectively represent ocean and land.**