# Peer review of "First Results of Cloud Retrieval from Geostationary Environmental Monitoring Spectrometer"

_Atmospheric Measurement Techniques, 2023_

## Author Comment (AC1)

**Manuscript ID:** amt-2023-91

**First Results of Cloud Retrieval from Geostationary Environmental Monitoring Spectrometer**

Bo-Ram Kim, Gyuyeon Kim, Minjeong Cho, Yong-Sang Choi[*], and Jhoon Kim

**Item-by-item responses to Reviewer 1's comments:**

We appreciated Reviewer 1's interest in our study and your valuable suggestion. We have carefully reviewed your comments and revised the manuscript as clearly as possible. We have highlighted the revised sections in blue in the manuscript. The revised manuscript has been proofread.

1) To improve the structure of this paper, it is necessary to revise the order of sections. Especially, Sections 2.2 and 3.1 are too similar. Please merge the two sections into one section.

[Reply] The contents of the two sections are in fact completely different. The only misleading aspect was the similar titles. In the revised manuscript, Section 2.1 (which was Section 2.2 in the original version) is now entitled "GEMS Cloud Algorithm Description," while Section 3.1 is now entitled "Satellite Data Used in This Study."

2) To explain the GEMS radiance characteristics, sampling and resolution information are wrong. The sampling is 0.2 nm and the resolution is 0.6 nm. Please correct it.

[Reply] We corrected to 0.6 nm.

3) Section 2.2 has to be moved before Section 2.1. Algorithm outputs are more appropriate after the algorithm description.

[Reply] The section is moved as suggested.

4) L139: For the radiance simulation, this study used 460-485 nm. However, the O2-O2 band at 477 nm significantly affect the spectral range longer than 485 nm. Why does this study use the spectral range with not wide enough?

[Reply] We have revised the manuscript in lines 120-122 as follows:

*"Before the launch of GEMS, we designed the algorithm using 460-490 nm. However, we discovered that a bad-pixels exist over 485 nm in partial area (Lee et al., 2023) during the in-orbit test of GEMS. Therefore, the fitting window was shortened for stable cloud retrieval."*

5) L144: This spectral range is significantly affected the NO2 absorption. Why does this study not consider the NO2 absorption?

[Reply] We decided the gases in cloud retrieval process based on fitting residual. In the fitting residual analysis, both of $NO_2$ and $O_3$ did not cause remarkable error, but only $O_3$ caused error in high cloud conditions. Therefore, we added the $O_3$ absorption effect.

We revised manuscript in lines 126-129 as follows:

*"In addition, even though $NO_2$ absorption coefficients exist in the spectral range of the input reflectance, their effects are disregarded because the impact is negligible."*

In addition, we acknowledged the effects of $NO_2$ absorption on the GEMS cloud retrieval algorithm, so we attempted to account for $NO_2$ absorption via the adoption of QDOAS, etc. We also included the related discussion in lines 426-427 as follows:

*"Also, QDOAS application enable to consider the $NO_2$ absorption. We anticipate the improvement of the GEMS cloud retrieval algorithm in the future through consider those remaining issues."*

6) L156: From Equation (3), this equation is not a full DOAS method. It is linearized absorption signal separation. The full DOAS method is additionally considered the non-linear correction. How about correcting the non-linear effect for the estimation of SCD?

[Reply] We tested the effect of adding a nonlinear term of the second and third order to Equation (3), but the results show no statistically significant difference when nonlinear terms are considered. The initial goal of the GEMS cloud was to produce results as stable as possible, so we aimed to reduce the uncertainty caused by inversion calculations using linear fitting. However, we conducted the fitting analysis using synthetic data, we agree that nonlinear fitting analysis is needed using operated GEMS L1C and IRR data. One approach for applying nonlinear fitting is to utilize QDOAS, as demonstrated by Danckaert et al. (2012). We plan to analyze the results of cloud calculations using nonlinear fitting in addition to the existing results in future research.

We included the related discussion in lines 423-426 as follows:

*"Since the GEMS cloud is retrieved through the DOAS method using linear fitting, it is necessary to compare the results with nonlinear fitting. One approach for applying nonlinear fitting is to utilize QDOAS, as demonstrated by Danckaert et al. (2012). We plan to analyze the results of cloud retrievals using the nonlinear fitting method in existing algorithm in future research."*

7) L161: For readability, please add the flowchart of the whole cloud retrieval algorithm.

[Reply] We added the flowchart of GEMS cloud retrieval algorithm in Fig. 1.

[Figure]

Figure 1: GEMS cloud retrieval algorithm flowchart.

565

8) L162-L166: It is too simple to explain. Please add the details of the explanation what is the main difference between the two different platforms.

[Reply] We revised manuscript in lines 139-143 as follows:

*"As already noted, the observational geometry differs from the LEO orbit because GEMS is in GEO. The VZA is constant at each location, but it varies significantly across the observation area. In contrast, SZA varies significantly across the complete observation area from sunrise to sunset. This change in SZA influences not only the calculation of reflectance, the optical path, etc., but also the variation in surface reflectance. Therefore, we simulated the RTM considering the various conditions of surface reflectance."*

9) Section 3.1 (before revision): This section needs to separate the sections according to the platforms.

[Reply] We separated the sections for each platform.

10) L248: This manuscript is only focused on the Korean Peninsula, not the GEMS domain. Do you have some reasons? Please clarify this issue.

[Reply] We considered the entire swath (from 03 to 06 UTC) within the GEMS FOV. We revised the manuscript in lines 248-251 as follows:

*"TROPOMI and CALIOP passed the GEMS FOV between 03 and 06 UTC, we use the entire swath to validate the GEMS products, but since most cases occurred close to the Korean Peninsula, we primarily used around the 04 UTC observation for validation. GEMS observes the*

*Korean Peninsula at ~ 45 minute past every hour during daytime, therefore, data observed at 04:45 UTC was selected as the validation data."*

11) Section 4: This manuscript is focused on the cloud algorithm for GEMS. However, Section 4 shows the results using OMI and TROPOMI. Please clarify the purpose of these results.

[Reply] Using the same input data is valuable for understanding algorithm differences without sensor error. We revised the manuscript in lines 268-270 as follows:

*"Using the same input data is valuable for understanding algorithm differences. Since OMI is the foundation of the GEMS cloud retrieval algorithm, the GEMS cloud algorithm would produce a product that is highly correlated with the OMI cloud product."*

12) Figure 3: This figure is one of the key results in this manuscript. However, as shown in this figure, an arbitral stripe pattern exists. I think that this is a problem during the inversion process for the best solution estimation from LUT. This problem is a critical issue for the operational algorithm. Please check and find the reason and need to fix this critical issue.

[Reply] The striped pattern caused by look-up table interpolation processes was eliminated after the systematic error was resolved as follows.

[Figure]

Figure 4: This figure is similar to Figure 1 but for TROPOMI. To verify the performance and seasonal dependency of GEMS cloud retrieval algorithm, cloud retrieval results were compared with those of TROPOMI for random days in March, May, September, and December 2021. The top panel shows the results for ECF and the bottom panel shows the results for CCP in a density scatterplot. As definitions of cloud pressure differ between GEMS and TROPOMI, the density scatterplot was generated using the mean value of CTP and CBP (mean cloud pressure, mCP) from TROPOMI. The x-axis shows the TROPOMI values, and the y-axis shows the GEMS cloud retrieval results. The solid line represents the 1:1 line, and the correlation coefficient and regression equation are also displayed.

13) Section 5: Although the manuscript shows several cases, the long-term comparison result is also required to see the stability of the algorithm's accuracy. In addition, this study only showed the 0430 UTC result. It is only allowed to the intercomparison. To see the diurnal variability of cloud retrieval results, this study also has to show the continuous diurnal results. In addition, some performance results (statistical validation results) are essential.

[Reply] The reason why we used only the 04:45 UTC data is TROPOMI passed over the Korean Peninsula at 04 UTC. We utilized approximately over 300,000 pixels each month. The validation

results for TROPOMI cloud products over the six-month period in 2023 are presented in the table below.

Table 1. Monthly validation results between GEMS and TROPOMI cloud product

| Mon | ECF | | CCP | |
|-----|-----|-----|-----|-----|
| Jan | R = 0.74 | Y = 0.77x+0.18 | R = 0.88 | Y = 0.54x+356.14 |
| Feb | R = 0.78 | Y = 0.81x+0.16 | R = 0.88 | Y = 0.54x+354.75 |
| Mar | R = 0.77 | Y = 0.79x+0.14 | R = 0.82 | Y = 0.48x+392.42 |
| Apr | R = 0.78 | Y = 0.79x+0.14 | R = 0.83 | Y = 0.51x+361.75 |
| May | R = 0.78 | Y = 0.79x+0.13 | R = 0.81 | Y = 0.49x+371.01 |
| Jun | R = 0.75 | Y = 0.76x+0.15 | R = 0.72 | Y = 0.43x+398.01 |

We have performed the error by temporal variations and added the paragraph in lines 356-361 as follows:

*"Due to the GEO orbit in which the GEMS is located, there are temporal variations in SZA as well as several input variables. Using the time-specific observation data of the aerosol high concentration case, we tested the error when input variables over time were not taken into account and were inaccurately entered as fixed values. The results indicate that the largest error in cloud calculation occurs when the SZA change is not adequately accounted for. The surface reflection, which causes an approximately 1% change in time, also caused a significant error, and the error characteristics in the ground-ocean were found to be substantially different. The error caused by a minor change in ground pressure over time was insignificant."*

We have confirmed the movement of typhoon over time and added the paragraph in lines 381-384 as follows:

*"Analysis of cloud production characteristics over time is required despite the fact that verification with other satellites is not feasible for all GEMS observation times except when some paths overlap. In the case of typhoons in which clouds are transparent and move rapidly over time, the observation of cloud movement over time was therefore qualitatively analyzed. It was discovered that GEMS cloud algorithm accurately identified the movement of typhoons over time."*

Minor Comment

1) L160: angle geometry → observation geometry

[Reply] We changed all expressions 'angle geometry' to 'observation geometry'.

2) L168: Please add the reference for the VLIDORT NGST version.

[Reply] We added the reference for the VLIDORT NGST version in lines 146 as follows:

*"while GEMS used the VLIDORT NGST version (Spurr et al., 2006)."*

3) L199: Please add the reference, and recheck the spatial resolution of TROPOMI in L204.

[Reply] We added references and revised the spatial resolution as $3.5 \times 5.5$ km$^2$ in lines 206-207 as follows:

*"Launched in 2017, TROPOMI is in SSO (as is OMI) and has a spatial resolution of 3.5 × 5.5 km, which is similar to that of GEMS (Latsch et al., 2022),"*

4) L241: To the clarify the difference, please list-up the difference of definition for cloud parameters.

[Reply] We added the table 2 to show the difference of definition for cloud parameters.

5) L285-286: This sentence is not clear. Please rephrase it.

[Reply] We rephrased the sentence in lines 290 as follows:

*"Figure 3 displays the outcome of the GEMS cloud process, derived from OMI observations conducted on March 25th, 2007."*

---

## Author Comment (AC2)

**Manuscript ID:** amt-2023-91

**First Results of Cloud Retrieval from Geostationary Environmental Monitoring Spectrometer**

Bo-Ram Kim, Gyuyeon Kim, Minjeong Cho, Yong-Sang Choi[*], and Jhoon Kim

**Item-by-item responses to Reviewer 2's comments:**

We appreciated Reviewer 2's interest in our study and your valuable suggestion. We have carefully reviewed your comments and revised the manuscript as clearly as possible. We have highlighted the revised sections in blue in the manuscript. The revised manuscript has been proofread.

General comments:

This manuscript presents a generally well written study on the algorithm retrieval results of GEMS cloud products. The author presented comprehensive analysis including comparison with the different satellite products along with the algorithm results. I suggest the publication of the paper after minor revisions.

Specific comments:

Line 11: 'the first geostationary orbit satellite' → I recommend that to be more specific as the GEMS is not the first geostationary orbit satellite instrument.

[Reply] We revised the words in lines 11 as follows:

*"the first environmental geostationary orbit satellite"*

Line 26: missing periods.

[Reply] We entered periods.

Line 23,29: It may not be a serious problem, but I suggest you distinguish the word between "satellite" and "instruments".

[Reply] We revised the satellite to instruments.

Line 30: gases(GHG) → gases (GHG)

[Reply] We put the space between words and parentheses.

Line 31: Please check typo "using use spectrometers GHGs".

[Reply] We revised the sentence in lines 30-31 as follows:

*"Satellites now monitor global warming by measuring greenhouse gases such as carbon dioxide and methane using spectrometers in the near-infrared and shortwave-infrared regions"*

Line 41: 'characteristics vary greatly depending on the spectral band.' → Do you have any reference or evidence for this sentence? Or did you want to say retrieval results greatly depending on the instrument characteristics?

[Reply] We intend that the characteristics of clouds are depending on the instrument characteristics.

Line 58: I thought the spectral resolution of GEMS is about 0.6nm, while sampling is 0.2nm. Could you check again?

[Reply] We revised the 0.2 nm to 0.6 nm.

Line 60: 'keeping the Sun-Earth-satellite angle constant' → do you mean constant VZA?

[Reply] Yes, we revised the words in lines 59 as follows:

*"in a sun-synchronous orbit (SSO) while maintaining the Solar Zenith Angle (SZA) variation"*

Line 64-65: Could you provide a reference for this? Or I think you can probably explain it with low SNR, etc.

[Reply] We added the reference and revised the manuscript in lines 62-63 as follows:

*"This reduces the quantity of radiation energy reaching the satellite and extends the beam path, resulting in a lower signal-to-noise ratio (Vandaele et al., 2018) and cloud retrieval errors."*

Line 84: 'with a resolution of 0.2 nm' → Could you check this again? I thought the spectral resolution of GEMS is about 0.6nm.

[Reply] We revised the 0.2 nm to 0.6 nm.

Line 92: Could you provide some references?

[Reply] Cloud phase is typically based on 8.7 micron and cloud top properties are based on thermal bands therefore, the cloud top heights from UV-VIS bands are less than those retrieved by thermal bands. I also added references: Compernolle et al., 2021 and Kim et al., 2019.

Line116: Why? Are there no CRF products from other satellites?

[Reply] The CRF is only converted from ECF considering wavelength dependency and it is significantly related to ECF, therefore we did not need more comparison for CRF. This is the reason why there are no CRF products from other satellites.

Line 144,147,152: Just curious. Why is there no consideration of $NO_2$ in this equation? I think the impact may be significant, especially over East Asia. What do you mean that the absorption by nitrogen dioxide is linear?

[Reply] We decided the gases in cloud retrieval process based on fitting residual. In the fitting residual analysis, both of $NO_2$ and $O_3$ did not cause remarkable error, but only $O_3$ caused error in high cloud conditions. Therefore, we added the $O_3$ absorption effect.

We apologize for the confusion caused by mistakenly describing it as linear, we deleted in manuscript in lines 127-129 as follows:

*"In addition, even though $NO_2$ absorption coefficients exist in the spectral range of the input reflectance, their effects are disregarded because the impact is negligible."*

In addition, we acknowledged the effects of $NO_2$ absorption on the GEMS cloud retrieval algorithm, so we attempted to account for $NO_2$ absorption via the adoption of QDOAS, etc. We also included the related discussion in lines 426-427 as follows:

*"Also, QDOAS application enable to consider the $NO_2$ absorption. We anticipate the improvement of the GEMS cloud retrieval algorithm in the future through consider those remaining issues."*

Line 168: Do you have any reference paper for VLIDORT NGST?

[Reply] We added the reference for the VLIORT NGST in lines 146 as follows:

*"while GEMS used the VLIDORT NGST version (Spurr et al., 2006)."*

Line 173: 'which has the most similar algorithm design' → Do you mean as a prototype?

[Reply] We revised the sentence in lines 175-176 as follows:

*"which has the most similar to spectral resolution and cloud prototype algorithm of GEMS"*

Line 174: 'operates simultaneously with GEMS' → I suggest 'in operational since 2018' rather than operates simultaneously.

[Reply] We revised the words following your suggestion.

Line 175: 'the same orbit' → Does it mean the same geostationary orbit?

[Reply] Yes, we revised it in lines 177 as follows:

*"the same GEO orbit "*

Line 182: 0.6 → 0.6nm

[Reply] We added the unit.

Line 243: Just curious. Are there any standards to select the cases?

[Reply] We choose the cases following some reasons. The case used to have occurred in GEMS FOV. And then we choose the specific day for each case have to be captured on all platforms. And the meaning of each case is as follows:

East Asia has experienced a severe problem with the high concentration of aerosol. Considering the purpose of GEMS to monitor the atmospheric environment, we need to validate a high-concentration case. The second case is for the typhoon, the typhoon is a very good case to validate cloud products because typhoon brings various cloud type. The last case is for sea fog, as I mentioned in the manuscript, sea fog is often between Korean Peninsula and China. Sea fog acts on the bright surface, therefore errors in their pressure cause very large errors in the gas retrievals. Therefore, we choose the sea fog cases and analyzed the cloud products.

Line 258: 'the nearest neighbor method was based on' → 'the nearest neighbor method was used based on'?

[Reply] We revised as suggested in lines 262.

Figure 1: Reason for the stripe pattern?

[Reply] The striped pattern caused by look-up table interpolation processes was eliminated after the systematic error was resolved as follows:

[Figure]

Figure 2: Cloud products for random days in each month of 2007 (classified by seasons: DJF-winter, MAM-spring, JJA-summer, SON-autumn) for comparison with cloud products from the OMI algorithm. The top panel shows the density scatterplot for effective cloud fraction (ECF) and the bottom panel shows the density scatter plot for cloud centroid pressure (CCP). The x-axis represents the values from OMI, and the y-axis represents the cloud products from the GEMS algorithm. The solid line indicates the 1:1 line, and the correlation coefficient (R), regression equation, root mean square error (RMSE), mean bias error (MBE), and number of pixels (#) used in the analysis are presented.

Line 280: It would be better if you could add the reason briefly.

[Reply] We revised the manuscript in lines 284-289 as follows:

*"However, many previous studies (Vasilkov et al., 2008; Sneep et al., 2008; Loyola et al., 2018; Compernolle et al., 2021) reported that the accuracy of CCP retrieval using O2-O2 absorption was significantly lower for areas with an ECF less than 0.2."*

Figure 2: Caption 'March 5th' → 'March 25th'.

[Reply] We revised the typo.

Figure 1-4: I can see that you are using the term "GEMS ECF" or "GEMS CCP". This is not GEMS data, but GEMS algorithm applied results. I suggest you distinguish between real GEMS products and GEMS algorithm applied products in the Figure, but I'll let the author decide it.

[Reply] We changed the expression to GEMS-applied ECF, GEMS-applied CCP

Figure 3: Reason for the stripe pattern?

[Reply] Same problem in Fig. 1 (original manuscript). The striped pattern caused by look-up table interpolation processes was eliminated after the systematic error was resolved as follows:

[Figure]

Figure 4: This figure is similar to Figure 1 but for TROPOMI. To verify the performance and seasonal dependency of GEMS cloud retrieval algorithm, cloud retrieval results were compared with those of TROPOMI for random days in March, May, September, and December 2021. The top panel shows the results for ECF and the bottom panel shows the results for CCP in a density scatterplot. As definitions of cloud pressure differ between GEMS and TROPOMI, the density scatterplot was generated using the mean value of CTP and CBP (mean cloud pressure, mCP) from TROPOMI. The x-axis shows the TROPOMI values, and the y-axis shows the GEMS cloud retrieval results. The solid line represents the 1:1 line, and the correlation coefficient and regression equation are also displayed.

Line 329: Just curious. Which channel does the AMI use for the cloud retrieval?

[Reply] The AMI uses the VIS (0.6 μm, 0.8 μm) and IR (10.4 μm) channels for cloud mask retrieval.

Line 350: Why does the GEMS tends to estimate lower cloud heights than TROPOMI cloud pressure?

[Reply] $O_2$-$O_2$ is related to the square of pressure and converged very low cloud, therefore it is sensitive to very low clouds (larger than 700 hPa, scale pressure is located around 700 hPa). We changed and added in the manuscript in lines 350-355 as follows:

*"Comparing TROPOMI CH with GEMS CH revealed that, in general, GEMS tends to concur with TROPOMI CH in low cloud (less than 6 km) conditions, but estimates lower cloud heights than TROPOMI cloud pressure calculations for high cloud (over 6 km). The scale pressure of O2-O2 absorption is approximately 700 hPa, and this altitude becomes the reference for the relationship between TROPOMI and GEMS cloud pressure. Consequently, while lower clouds*

*display cloud heights comparable to TROPOMI, clouds at higher altitudes have the characteristic of estimating lower altitudes."*

Line 367: Again, why does the GEMS tends to estimate lower cloud heights than TROPOMI cloud pressure?

[Reply] GEMS tended to underestimate in high and thick cloud conditions, therefore, we added in the manuscript in lines 377 as follows:

*"GEMS tended to underestimate cloud height compared with TROPOMI in high and thick cloud conditions."*

Line 398: Maybe you can add a brief reason why GEMS cloud height is the lowest.

[Reply] We revised the manuscript in lines 409-411 as follows:

*"The GEMS cloud retrieval algorithm is sensitive to high pressure greater than scale pressure; as a result, GEMS estimates of cloud height were the lowest among the four satellites, corresponding to the height at which clouds reflect radiation."*

---

## Author Response (AR2)

**Manuscript ID:** amt-2023-91

**First Results of Cloud Retrieval from Geostationary Environmental Monitoring Spectrometer**

Bo-Ram Kim, Gyuyeon Kim, Minjeong Cho, Yong-Sang Choi[*], and Jhoon Kim

**Item-by-item responses to Reviewer 1's comments:**

We appreciated Reviewer 1's interest in our work as well as your insightful recommendation. We have carefully considered your suggestions and made improvements to the manuscript. We have kept track of the manuscript's updated sections. Proofreading has been done on the updated manuscript.

1) L19-L32: This paragraph is too broad to include the manuscript. Although the cloud is important for the radiation field, the manuscript will be merely focused on cloud retrieval.

[Reply] We have revised the first paragraph to include only essential information regarding weather satellite observations and cloud issues, incorporating the reviewer's feedback. Additionally, we have structured the second paragraph to provide more detailed insights on cloud effects. The updated manuscript in L19-L35 as follows:

*Atmospheric composition has been monitored continuously by several satellite-loaded instruments since 1978: Total Ozone Mapping Spectrometer (TOMS), the Ozone Monitoring Instrument (OMI), Global Ozone Monitoring Experiment (GOME), SCanning Imaging Absorption spectroMeter for Atmospheric CHartographY (SCIAMACHY), and Tropospheric Monitoring Instrument (TROPOMI) (Hsu et al., 1997; Burrows et al., 1999; Bovensmann et al., 1999; Levelt et al., 2006; Veefkind et al., 2012). These spectrometers measure the ultraviolet (UV) and visible (Vis) radiation centered at 240 to 790 nm (Hsu et al., 1997; Burrows et al., 1999; Bovensmann et al., 1999; Levelt et al., 2006; Veefkind et al., 2012). It is then required to estimate the beam path length of the radiation to retrieve precise atmospheric compositions from the radiation measured by these spectrometers. The beam path length of the radiation is the entire path length of incoming and reflected solar energy by Earth's surface until reaching the satellite. Thus the calculation requires to consider geometric factors such as solar zenith angle (SZA) and viewing zenith angle (VZA).*

*The beam path length should be also calculated not only for a clear-sky condition, but also for a cloudy-sky condition. This is because cloud layers can shorten the beam path length by blocking the beam from atmospheric components below the clouds. Cloud reflectance is typically greater than that of most surfaces (excluding snow and ice) and this cloud effect can inevitably result in significant errors in the observations of atmospheric variables (Hong et al., 2017; Chimot et al., 2018). Therefore, to obtain accurate concentrations of atmospheric components, it is necessary to evaluate and quantify the cloud effects on the beam path length.*

2) L163-L167: I am confused as to why the cloud models (LER and MLER) are explained. During the retrieval, the used points for LER and MLER have to be clarified.

[Reply] "MLER" stands for Mixed LER model. In this study, we assumed the reflectance characteristics of clouds to be LER and set the albedo to 0.8, to valuate cloud reflectance values

for each pixel. Therefore, it can be considered that we used the MLER cloud model. Please refer to L95 for the explanation of the abbreviation "MLER", as follows: *"the mixed Lambertian equivalence reflectivity (MLER) cloud model"*

3) Section 3.1: For the readability, all used satellite sensors' specifications are listed as Table.

[Reply] We added the sensor's specification in Table 2:

**Table 1: Overview of the cloud products included in this study.**

| Instrument | Spectral range (nm) | Cloud product name | Variable | Cloud spectral range (nm) |
|---|---|---|---|---|
| OMI | 270–314, 306–380, 350–500 | OMCLDO2 | Effective cloud fraction
Cloud height | 460–490 |
| TROPOMI | 270–495, 710–775, 2305–2385 | ROCINN CRB | Cloud fraction
Cloud albedo
Cloud pressure/height | 758–766 |
| GEMS | 300–500 | GEMS CLD | Effective cloud fraction
Cloud centroid pressure | 460–485 |
| AMI | 470, 511, 640, 856, 1380,
1610, 3830, 6241, 6952,
7344, 8592, 9625, 10403,
11212, 12364, 13310 | GK2A CTH | Cloud top height | 8592–13310 |
| CALIOP | 532, 1064 | VFM | Vertical feature mask | 532, 1064 |

4) L196-L200: The adopted method is too confused. Why did the GEMS cloud retrieval algorithm adopt to the OMI radiance? The purpose and details of methodology is needed.

[Reply] Since GEMS cloud retrieval algorithm is based on OMI's cloud retrieval algorithm, we aimed to evaluate the performance of the GEMS cloud retrieval algorithm by comparing it to OMI level 2 cloud products for validation first. We used OMI's level 1B data as proxy data for the GEMS cloud retrieval method in order to optimize consistency between the two algorithms' experimental settings. The purpose and details of methodology are clarified in the beginning of Chapter 3, as follows: *"To evaluate the performance of the GEMS cloud retrieval algorithm, cloud products were produced using OMI radiance data, which are similar to the GEMS spectral resolution and cloud prototype algorithm."*

5) L221: Why L2-VFM data of CALIOP was used? The L2-VFM only show the existence of cloud layer, not a profile.

[Reply] CALIOP is the only active sensor satellite operated simultaneously with GEMS, and while it may not provide information on low-level clouds in thick cloud cover, it can be regarded as the instrument that most accurately detects the presence of clouds. As the definitions of the cloud height products were all different depending on satellite instruments, we utilized CALIOP's

VFM data which can show the vertical presence of clouds, in our comparisons between GEMS and CALIOP. The expression "cloud-aerosol profile" has been modified to *"cloud-aerosol vertical existence"* in L213.

6) Section 5: Not only the long-term retrieval results used by non-GEMS sensors' radiance, the author also has to evaluate the long-term retrieval performance of cloud retrieval algorithm using GEMS radiance. Please add the long-term performance result of GEMS cloud algorithm using GEMS Level 1 radiance.

[Reply] Thank you for your suggestion. We added *Section 5.4 Monthly Cloud Product Validation* and Figure 12 to present the evaluation result for the long-term retrieval performance as follows:

*5.4 Monthly Cloud Product Validation*

*For the monthly validation of GEMS cloud products, we randomly selected 3 days of each month from 2021 to 2022 and conducted the validation against TROPOMI data. To exclude the influence of variations in GEMS observation areas due to changing seasons, we employed the full west mode and selected the times when TROPOMI observation paths were present for the validation. Collocation was performed using the same method as described in Section 4.2, to assess the cloud products from both satellites. For certain periods, TROPOMI provides cloud products in both the OFFL (Offline) and RPRO (Reprocessed) versions, so we presented the correlation coefficients from the validation using both products (Fig. 12). In addition, we accounted for land cover since the precision of cloud product retrievals can vary between water and land due to factors like surface reflectance, as indicated by the results of scene analyses.*

*For 2 years of monthly validation results, in the case of ECF, there appeared to be no significant monthly variations in accuracy where higher accuracy was observed over ocean as compared to land, in general. Furthermore, the difference in validation results based on TROPOMI versions was not pronounced. On the other hand, for CCP, substantial monthly variations in accuracy were observed, especially a noticeable decrease in CCP correlation coefficients during the summer seasons (June, July, August) over ocean. Additionally, variations in accuracy were evident depending on TROPOMI versions, with the newly provided RPRO version showing improved correlation with GEMS.*

*The difference in ECF accuracy based on land cover can largely be attributed to the use of OMI climatology values for surface reflectance as input data.   It is expected that this accuracy difference between the land and water based on land cover will significantly decrease when surface reflectance data observed by GEMS is applied as inputs.*

[Figure]

*Figure 1: The monthly correlation coefficient (R) values between GEMS cloud products and TROPOMI OFFL version (solid line) and RPRO version (dotted line) are presented in Figures (a) for ECF and (b) for CCP. The red and blue lines respectively represent water and land.*